# LEARNING TO STEER MARKOVIAN AGENTS UNDER MODEL UNCERTAINTY

**Jiawei Huang[†], Vinzenz Thoma[‡], Zebang Shen[†], Heinrich H. Nax[§], Niao He[†]**
† Department of Computer Science, ETH Zurich
`{jiawei.huang, zebang.shen, niao.he}@inf.ethz.ch`
‡ ETH AI Center
`vinzenz.thoma@ai.ethz.ch`
§ University of Zurich
`heinrich.nax@uzh.ch`

## ABSTRACT

Designing incentives for an adapting population is a ubiquitous problem in a wide array of economic applications and beyond. In this work, we study how to design additional rewards to steer multi-agent systems towards desired policies *without* prior knowledge of the agents' underlying learning dynamics. Motivated by the limitation of existing works, we consider a new and general category of learning dynamics called *Markovian agents*. We introduce a model-based non-episodic Reinforcement Learning (RL) formulation for our steering problem. Importantly, we focus on learning a *history-dependent* steering strategy to handle the inherent model uncertainty about the agents' learning dynamics. We introduce a novel objective function to encode the desiderata of achieving a good steering outcome with reasonable cost. Theoretically, we identify conditions for the existence of steering strategies to guide agents to the desired policies. Complementing our theoretical contributions, we provide empirical algorithms to approximately solve our objective, which effectively tackles the challenge in learning history-dependent strategies. We demonstrate the efficacy of our algorithms through empirical evaluations.

## 1 INTRODUCTION

Many real-world applications can be formulated as Markov Games (Littman, 1994) where the agents repeatedly interact and update their policies based on the received feedback. In this context, different learning dynamics and their convergence properties have been studied extensively (see, for example, Fudenberg and Levine (1998)). Because of the mismatch between the individual short-run and collective long-run incentives, or the lack of coordination in decentralized systems, agents following standard learning dynamics may not converge to outcomes that are desirable from a system designer perspective, such as the Nash Equilibria (NE) with the largest social welfare. An interesting class of games that exemplify these issues are so-called "Stag Hunt" games (see Fig. 1-(a)), which are used to study a broad array of real-world applications including collective action, public good provision, social dilemma, team work and innovation adoption (Skyrms, 2004)[1]. Stag Hunt games have two pure-strategy NE, one of which is 'payoff-dominant', that is, both players obtain higher payoffs in that equilibrium than in the other. Typical algorithms may fail to reach the payoff-dominant equilibrium $(H, H)$ (LHS Fig. 1-(b)). Indeed, the other equilibrium $(G, G)$ is typically selected when it is risk-dominant (Harsanyi and Selten, 1988; Newton, 2021).

This paper focuses on situations when an external "mediator" exists, who can influence and *steer* the agents' learning dynamics by modifying the original rewards via additional incentives. This kind of mediator can be conceptualized in various ways. In particular, we can think of a social planner who provides monetary incentives for joint ventures or for adoption of an innovative technology via individual financial subsidies. As illustrated on the RHS of Fig. 1-(b), with suitable steering, agents' dynamics can be directed to the best outcome. Our primary objective is to steer the agents to some desired policies, that is, to minimize the **steering gap** vis-a-vis the target outcome. As a secondary

---

[1]We defer a concrete and practical scenario which can be modeled by the Stag Hunt game to Appx. B.1

objective, the payments to agents regarding the steering rewards should be reasonable, that is, the ***steering cost*** should be low.

To our knowledge, Zhang et al. (2023) is the first work studying a similar steering problem as ours. They assume the agents are no-regret learners and may act in arbitrarily adversarial ways. In some natural settings, such assumption may not be practical, because no-regret criterion typically requires careful processing of the entire history of learning. In settings with limited cognitive resources and bounded rationality, it is natural to favor models where the agents only process a subset of the available information (Camerer, 2011). In particular, humans have been widely shown to rely overproportionally on recent experiences in decision making, known as 'recency bias' (Costabile and Klein, 2005; Page and Page, 2010; Durand et al., 2021). Besides, there is evidence that behavioral dynamics that only rely on the most recent experience are able to fit behavioral data well in certain situations (Mäs and Nax, 2016). Motivated by these insights, we therefore study steering a different category of learning dynamics called ***Markovian agents***, where the agents' policy updates only depend on their current policy and the (modified) reward function. Our model complements

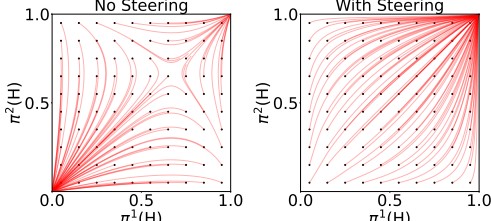

(a) Payoff Matrix of the Two-Player "Stag Hunt" Game. H and G stand for two actions `Hunt` and `Gather`. Both (H, H) and (G, G) are NE and (H, H) is payoff-dominant.

(b) Dynamics of Agents Policies without/with Steering. Agents follow natural policy gradient (replicator dynamics) for policy update. $x$ and $y$ axes correspond to the probability to take action H by the row and column players. Red curves represent the dynamics of agents' policies starting from different intializations (black dots).

Figure 1: Example: The "Stag Hunt" Game

the prior work on no-regret agents, and serves as the first abstraction of behavior based on limited cognitive abilities with recency bias in steering setting. Theoretically, Markovian agents subsumes a broader class of popular policy-based methods as concrete examples (Giannou et al., 2022; Ding et al., 2022), which are not covered by no-regret assumptions. We note that a concurrent work (Canyakmaz et al., 2024) considers a similar setting as ours, and we defer to Sec. 1.1 for further discussion.

In practice, learning the right steering strategies encounters two main challenges. First, the agents may not disclose their learning dynamics model to the mediator. As a result, this creates fundamental model uncertainty, which we will tackle with appropriate Reinforcement Learning (RL) techniques to trading-off exploration and exploitation. Second, it may be unrealistic to assume that the mediator is able to force the agents to "reset" their policies in order to generate multiple steering episodes with the same initial state. This precludes the possibility of learning steering strategies through episodic trial-and-error. Therefore, the most commonly-considered, fixed-horizon episodic RL (Dann and Brunskill, 2015) framework is not applicable here. Instead, we will consider a *finite-horizon non-episodic* setup, where the mediator can only generate one finite-horizon episode, in which we have to conduct both the model learning and steering of the agents simultaneously. Motivated by these considerations, we would like to address the following question in this paper:

> *How to learn desired steering strategies for Markovian agents*
> *in the non-episodic setup under model uncertainty?*

We consider a model-based setting where the mediator can get access to a model class $\mathcal{F}$ containing the agents' true learning dynamics $f^*$. We summarize and highlight our key contributions as follow:

- **Conceptual Contributions**: In Sec. 3, we formulate steering as a non-episodic RL problem, and propose a novel optimization objective in Obj. (1), where we explicitly tackle the inherent model uncertainty by learning *history-dependent* steering strategies. As we show in Prop. 3.3, under certain conditions, even *without prior knowledge of $f^*$*, the optimal solution to Obj. (1) achieves not only low steering gap, but also "Pareto Optimality" in terms of both steering costs and gaps.

- **Theoretical Contributions**: In Sec. 4, we provide sufficient conditions under which there exists steering strategies achieving low steering gap. These results in turn justify our chosen objective and problem formulation.

- **Algorithmic Contributions**: Learning a history-dependent strategy presents challenges due to the exponential growth in the history space. We propose algorithms to overcome these issues.

  - When the model class $|\mathcal{F}|$ is small, in Sec. 5.1, we approach our objective from the perspective of learning in a Partially Observable MDP, and propose to to learn a policy over the model belief state space instead of over the history space.

  - For the case when $|\mathcal{F}|$ is large, exactly solving Obj. (1) can be challenging. Instead, we focus on approximate solutions to trade-off optimality and tractability. In Sec. 5.2, we propose a First-Explore-Then-Exploit (FETE) framework. Under some conditions, we can still ensure the directed agents converge to the desired outcome.

- **Empirical Validation**: In Sec. 6, we evaluate our algorithms in various representative environments, and demonstrate their effectiveness under model uncertainty.

## 1.1 CLOSELY RELATED WORKS

We discuss the works most closely related to ours in this section, and defer the others to Appx. B.2.

**Steering Learning Dynamics** As mentioned in the introduction, Zhang et al. (2023) are the first to introduce the "steering problem", but their setting differs quite fundamentally from ours in several key aspects. Firstly, they assume that agents behave as no-regret and arbitrarily adversarial learners, which may be unrealistic in settings with limited information and feedback, and owed to agents' limited cognitive resources (Camerer, 2011) including recency bias (Costabile and Klein, 2005). Motivated by this, we instead focus on a broad class of Markovian dynamics. Secondly, the mediator's objective in Zhang et al. (2023) is to steer agents such that the average policy converges to the target NE while maintaining a sublinear accumulative budget, motivated by their infinite-horizon setup. In contrast, we consider the finite-horizon setting, and therefore, we are concerned with minimizing the steering gap of the terminal policy and the cumulative steering cost. Thirdly, when the desired NE is not pure, Zhang et al. (2023) require the mediator to be able to "give advice" to the players to facilitate coordination, while we do not allow the mediator to do this. Because of these differences, the methods and results obtained by Zhang et al. (2023) and us *are not directly comparable*, yet complement one another depending on application considered.

Perhaps the closest to ours is a concurrent work by Canyakmaz et al. (2024). They consider a similar finite-horizon non-episodic setup as ours, and experimentally investigate the use of control methods to direct game dynamics towards desired outcomes, in particular allowing for model uncertainty. Compared to their work, we handle the model uncertainty in a more principled way by employing *history-dependent* steering strategies, since history can serve as sufficient information set for decision making under uncertainty. This leads to several differences in the design principles between our algorithms and Canyakmaz et al. (2024). Concretely, we propose a learning objective for history-dependent strategies in Obj. (1), and two algorithms for low uncertainty (small $\mathcal{F}$) and high uncertainty (large $\mathcal{F}$) settings, respectively. In the former case, we contribute a belief-state based algorithm that can exactly solve Obj. (1), offering a stronger solution due to the theoretical guarantee in Prop. 3.3. For the latter, although both our FETE and SIAR-MPC (Canyakmaz et al., 2024) share a two-phase (exploration + exploitation) structure, ours represents a more general framework with a more advanced exploration strategy (see more explanation in Sec. 5.2). Besides, we develop additional novel theory regarding the existence of strategies with low steering gap.

## 2 PRELIMINARY

In the following, we formally define the finite-horizon Markov Game that we will focus on. We summarize all the frequently used notations in this paper in Appx. A.

**Finite Horizon Markov Game** A finite-horizon $N$-player Markov Game is defined by a tuple $G := \{\mathcal{N}, s_1, H, \mathcal{S}, \mathcal{A} := \{\mathcal{A}^n\}_{n=1}^N, \mathbb{P}, \boldsymbol{r} := \{r^n\}_{n=1}^N\}$, where $\mathcal{N} := \{1, 2, ..., N\}$ is the indices of agents, $s_1$ is the fixed initial state, $H$ is the horizon length, $\mathcal{S}$ is the finite shared state space, $\mathcal{A}^n$ is the finite action space for agent $n$, and $\mathcal{A}$ denotes the joint action space. Besides, $\mathbb{P} := \{\mathbb{P}_h\}_{h \in [H]}$ with $\mathbb{P}_h : \mathcal{S} \times \mathcal{A} \to \Delta(\mathcal{S})$ denotes the transition function of the shared state, and $r^n := \{r_h^n\}_{h \in [H]}$ with $r_h^n : \mathcal{S} \times \mathcal{A} \to [0, 1]$ denotes the reward function for agent $n$. For each agent $n$, we consider the non-stationary Markovian policies $\Pi^n := \{\pi^n = \{\pi_1^n, ..., \pi_H^n\} | \forall h \in [H], \pi_h^n : \mathcal{S} \to \Delta(\mathcal{A}^n)\}$. We denote

$\Pi := \Pi^1 \times ... \times \Pi^N$ to be the joint policy space of all agents. Given a policy $\boldsymbol{\pi} := \{\pi^1, ..., \pi^N\} \in \Pi$, a trajectory is generated by: $\forall h \in [H], \forall n \in [N], \quad a_h^n \sim \pi^n(\cdot|s_h), \ r_h^n \leftarrow r_h^n(s_h, \boldsymbol{a}_h), \ s_{h+1} \sim \mathbb{P}_h(\cdot|s_h, \boldsymbol{a}_h)$, where $\boldsymbol{a}_h := \{a_h^n\}_{n \in [N]}$ denotes the collection of all actions. Given a policy $\boldsymbol{\pi}$, we define the value functions by: $Q_{h|\boldsymbol{r}}^{n,\boldsymbol{\pi}}(\cdot, \cdot) := \mathbb{E}_{\boldsymbol{\pi}}[\sum_{h'=h}^{H} r_{h'}^n(s_{h'}, \boldsymbol{a}_{h'})|s_h = \cdot, \boldsymbol{a}_h = \cdot], \ V_{h|\boldsymbol{r}}^{n,\boldsymbol{\pi}}(\cdot) := \mathbb{E}_{\boldsymbol{\pi}}[\sum_{h'=h}^{H} r_{h'}^n(s_{h'}, \boldsymbol{a}_{h'})|s_h = \cdot]$, where we use $|r$ to specify the reward function associated with the value functions. In the rest of the paper, we denote $A_{|\boldsymbol{r}}^{n,\boldsymbol{\pi}} = Q_{|\boldsymbol{r}}^{n,\boldsymbol{\pi}} - V_{|\boldsymbol{r}}^{n,\boldsymbol{\pi}}$ to be the advantage value function, and denote $J_{|\boldsymbol{r}}^n(\boldsymbol{\pi}) := V_{1|\boldsymbol{r}}^{n,\boldsymbol{\pi}}(s_1)$ to be the total return of agent $n$ w.r.t. policy $\boldsymbol{\pi}$.

# 3 THE PROBLEM FORMULATION OF THE STEERING MARKOVIAN AGENTS

We first introduce our definition of Markovian agents. Informally, the policy updates of Markovian agents are independent of the interaction history conditioning on their current policy and observed rewards. This encompass a broader class of popular policy-based methods as concrete examples (Giannou et al., 2022; Ding et al., 2022; Xiao, 2022; Daskalakis et al., 2020).

**Definition 3.1** (Markovian Agents). Given a game $G$, a finite and fixed $T$, the agents are Markovian if their policy update rule $f$ only depends on the current policy $\boldsymbol{\pi}_t$ and the reward function $\boldsymbol{r}$:

$$\forall t \in [T], \quad \boldsymbol{\pi}_{t+1} \sim f(\cdot|\boldsymbol{\pi}_t, \boldsymbol{r}).$$

Here we only highlight the dependence on $\boldsymbol{\pi}_t$ and $\boldsymbol{r}$, and omit other dependence (e.g. the transition function of $G$). It is worth to note that we do not restrict whether the updates of agents' policies are independent or correlated with each other, deterministic or stochastic. We assume $T$ is known to us.

In the steering problem, the mediator has the ability to change the reward function $\boldsymbol{r}$ via the steering reward $\boldsymbol{u}$, so that the agents' dynamics are modified to:

$$\forall t \in [T], \quad \boldsymbol{u}_t \sim \psi_t(\cdot|\boldsymbol{\pi}_1, \boldsymbol{u}_1, ..., \boldsymbol{\pi}_{t-1}, \boldsymbol{u}_{t-1}, \boldsymbol{\pi}_t), \quad \boldsymbol{\pi}_{t+1} \sim f(\cdot|\boldsymbol{\pi}_t, \boldsymbol{r} + \boldsymbol{u}_t),$$

Here $\psi := \{\psi_t\}_{t \in [T]}$ denotes the mediator's "steering strategy" to generate $u_t$. We consider history-dependent strategies to handle the model uncertainty, which we will explain later. Besides, $\boldsymbol{u}_t := \{u_{t,h}^n\}_{h \in [H], n \in [N]}$, where $u_{t,h}^n : \mathcal{S} \times \mathcal{A} \to [0, U_{\max}]$ is the steering reward for agent $n$ at game horizon $h$ and steering step $t$. $U_{\max} < +\infty$ denotes the upper bound for the steering reward. With a bit of abuse of notation, here we treat both $\boldsymbol{r}$ and $\boldsymbol{u}_t$ as vectors with length $HN|\mathcal{S}||\mathcal{A}|$ and use $\boldsymbol{r} + \boldsymbol{u}_t$ to denote elementwise addition. For practical concerns, we follow the standard constraints that the steering rewards are non-negative.

The mediator has a terminal reward function $\eta^{\text{goal}}$ and a cost function $\eta^{\text{cost}}$. First, $\eta^{\text{goal}} : \Pi \to [0, \eta_{\max}]$ assesses whether the final policy $\boldsymbol{\pi}_{T+1}$ aligns with desired behaviors—this encapsulates our primary goal of a low steering gap. Note that we consider the general setting and do not restrict the maximizer of $\eta^{\text{goal}}$ to be a Nash Equilibrium. For instance, to steer the agents to a desired policy $\boldsymbol{\pi}^*$, we could choose $\eta^{\text{goal}}(\boldsymbol{\pi}) := -\|\boldsymbol{\pi} - \boldsymbol{\pi}^*\|_2$. Alternatively, in scenarios focusing on maximizing utility, $\eta^{\text{goal}}(\boldsymbol{\pi})$ could be defined as the total utility $\sum_{n \in [N]} J_{|\boldsymbol{r}}^n(\boldsymbol{\pi})$. For $\eta^{\text{cost}} : \Pi \to \mathbb{R}_{\geqslant 0}$, it is used to quantify the steering cost incurred while steering. In this paper, we fix $\eta^{\text{cost}}(\boldsymbol{\pi}, \boldsymbol{u}) := \sum_{n \in [N]} J_{|\boldsymbol{u}}^n(\pi^n)$ to be the total return related to $\boldsymbol{\pi}$ and the steering reward $\boldsymbol{u}$. Note that we always have $0 \leqslant \eta^{\text{cost}}(\boldsymbol{\pi}, \boldsymbol{u}) \leqslant U_{\max} NH$.

**Steering Dynamics as a Markov Decision Process (MDP)** Given a game $G$, the agents' dynamics $f$ and $(\eta^{\text{cost}}, \eta^{\text{goal}})$, the *steering dynamics* can be modeled by a finite-horizon MDP. $M := \{\boldsymbol{\pi}_1, T, \Pi, \mathcal{U}, f, (\eta^{\text{cost}}, \eta^{\text{goal}})\}$ with initial state $\boldsymbol{\pi}_1$, horizon length $T$, state space $\Pi$, action space $\mathcal{U} := [0, U_{\max}]^{HN|\mathcal{S}||\mathcal{A}|}$, stationary transition $f$, running reward $\eta^{\text{cost}}$ and terminal reward $\eta^{\text{goal}}$. For completeness, we defer to Appx. B.3 for an introduction of finite-horizon MDP

**Steering under Model Uncertainty** In practice, the mediator may not have precise knowledge of agents learning dynamics model, and the uncertainty should be taken into account. We will only focus on handling the uncertainty in agents' dynamics $f$, and assume the mediator has the full knowledge of $G$ and the reward functions $\eta^{\text{goal}}$ and $\eta^{\text{cost}}$. We consider the model-based setting where the mediator only has access to a finite model class $\mathcal{F}$ ($|\mathcal{F}| < +\infty$) satisfying the following assumption:

**Assumption A** (Realizability). The true learning dynamics $f^*$ is realizable, i.e. $f^* \in \mathcal{F}$.

**A Finite-Hoziron Non-Episodic Setup and Motivation** As motivated previously, we formulate steering as a finite-horizon non-episodic RL problem. To our knowledge, in contrast to our finite-horizon setting, most of the non-episodic RL settings consider the infinite-horizon setup with stationary or non-stationary transitions, and therefore, they are also not suitable here. We provide more discussion in Appx. B.2.

**Definition 3.2** (Finite Horizon Non-Episodic Steering Setting)**.** The mediator can only interact with *the real agents* for *one episode* $\{\boldsymbol{\pi}_1, \boldsymbol{u}_1, ..., \boldsymbol{\pi}_T, \boldsymbol{u}_T, \boldsymbol{\pi}_{T+1}\}$, where $\boldsymbol{\pi}_{t+1} \sim f^*(\cdot|\boldsymbol{\pi}_t, \boldsymbol{r} + \boldsymbol{u}_t)$ for all $t \in [T]$. Nonetheless, the mediator can get access to the simulators for all models in $\mathcal{F}$, and it can sample *arbitrary* trajectories and do episodic learning with those simulators to decide the best steering actions $\boldsymbol{u}_1, \boldsymbol{u}_2, ..., \boldsymbol{u}_T$ to deploy.

**The Learning Objective** Motivated by the model-based non-episodic setup, we propose the following objective function, where we search over the set of all history-dependent strategies, denoted by $\Psi$, to optimize the average performance over all $f \in \mathcal{F}$.

$$\psi^* \leftarrow \arg\max_{\psi \in \Psi} \frac{1}{|\mathcal{F}|} \sum_{f \in \mathcal{F}} \mathbb{E}_{\psi,f}\Big[\beta \cdot \eta^{\text{goal}}(\boldsymbol{\pi}_{T+1}) - \sum_{t=1}^{T} \eta^{\text{cost}}(\boldsymbol{\pi}_t, \boldsymbol{u}_t)\Big], \tag{1}$$

Here we use $\mathbb{E}_{\psi,f}[\cdot] := \mathbb{E}[\cdot|\forall t \in [T], \boldsymbol{u}_t \sim \psi_t(\cdot|\{\boldsymbol{\pi}_{t'}, \boldsymbol{u}_{t'}\}_{t'=1}^{t-1}, \boldsymbol{\pi}_t), \boldsymbol{\pi}_{t+1} \sim f(\cdot|\boldsymbol{\pi}_t, \boldsymbol{r} + \boldsymbol{u}_t)]$ to denote the expectation over trajectories generated by $\psi$ and $f \in \mathcal{F}$; $\beta > 0$ is a regularization factor. Next, we explain the rationale to consider history-dependent strategies. As introduced in Def. 3.2, we only intereact with the real agents once. Therefore, the mediator needs to use the interaction history with $f^*$ to decide the appropriate steering rewards to deploy, since the history is the sufficient information set including all the information regarding $f^*$ availale to the mediator.

We want to clarify that in our steering framework, we will first solve Obj. (1), and then deploy $\psi^*$ to steer real agents. The learning and optimization of $\psi^*$ in Obj. (1) only utilizes simulators of $\mathcal{F}$. Besides, after deploying $\psi^*$ to real agents, we will not update $\psi^*$ with the data generated during the interaction with real agents. This is seemingly different from common online learning algorithms which conduct the learning and interaction repeatedly(Dann and Brunskill, 2015). But we want to highlight that, given the fact that $\psi^*$ is history-dependent, it is already encoded in $\psi^*$ how to make decisions (or say, learning) in the face of uncertainty after gathering data from real agents. In other words, one can interpret that, in Obj. (1), the $\psi^*$ to be optimized is an "online algorithm" which can "smartly" decide the next steering reward to deploy given the past interaction history. As we will justify in the following, our Obj. (1) can indeed successfully handle the model uncertainty.

**Justification for Objective** (1) We use $C_{\psi,T}(f) := \mathbb{E}_{\psi,f}[\sum_{t=1}^{T} \eta^{\text{cost}}(\boldsymbol{\pi}_t, \boldsymbol{u}_t)]$ and $\Delta_{\psi,T}(f) := \mathbb{E}_{\psi,f}[\max_{\boldsymbol{\pi}} \eta^{\text{goal}}(\boldsymbol{\pi}) - \eta^{\text{goal}}(\boldsymbol{\pi}_{T+1})]$ as short notes of the steering cost and the steering gap (of the terminal policy $\boldsymbol{\pi}_{T+1}$), respectively. Besides, we denote $\Psi^\varepsilon := \{\psi \in \Psi | \max_{f \in \mathcal{F}} \Delta_{\psi,T}(\mathcal{F}) \leqslant \varepsilon\}^2$ to be the collection of all steering strategies with $\varepsilon$-steering gap. Based on these notations, we introduce two desiderata, and show how an optimal solution $\psi^*$ of Obj. (1) can achieve them.

**Desideratum 1** ($\varepsilon$-Steering Gap)**.** We say $\psi$ has $\varepsilon$-steering gap, if $\max_{f \in \mathcal{F}} \Delta_{\psi,T}(f) \leqslant \varepsilon$.

**Desideratum 2** (Pareto Optimality)**.** We say $\psi$ is Pareto Optimal if there does not exist another $\psi' \in \Psi$, such that (1) $\forall f \in \mathcal{F}, C_{\psi',T}(f) \leqslant C_{\psi,T}(f)$ and $\Delta_{\psi',T}(f) \leqslant \Delta_{\psi,T}(f)$; (2) $\exists f' \in \mathcal{F}$, s.t. either $C_{\psi',T}(f') < C_{\psi,T}(f')$ or $\Delta_{\psi',T}(f') < \Delta_{\psi,T}(f')$.

**Proposition 3.3.** *[Justification for Obj.* (1)*] By solving Obj.* (1)*: (1) $\psi^*$ is Pareto Optimal; (2) Given any $\varepsilon, \varepsilon' > 0$, if $\Psi^{\varepsilon/|\mathcal{F}|} \neq \varnothing$ and $\beta \geqslant \frac{U_{\max} N H T |\mathcal{F}|}{\varepsilon'}$, we have $\psi^* \in \Psi^{\varepsilon+\varepsilon'}$;*

Next, we give some interpretation. As our primary desideratum, we expect the agents converge to some desired policy that maximizes the goal function $\eta^{\text{goal}}$ after being steered for $T$ steps, regardless of the true model $f^*$. Therefore, we restrict the worst case steering gap to be small. As stated in Prop. 3.3, for any accuracy level $\varepsilon > 0$, as long as $\varepsilon/|\mathcal{F}|$-steering gap is achievable, by choosing $\beta$ large enough, we can approximately guarantee $\psi^*$ has $\varepsilon$-steering gap. For the steering cost, although it is not our primary objective, Prop. 3.3 states that at least we can guarantee the Pareto Optimality: competing with $\psi^*$, there does not exist another $\psi'$, which can improve either the steering cost or gap for some $f' \in \mathcal{F}$ without deteriorating any others.

---

[2] In fact, besides $\varepsilon$, $\Psi^\varepsilon$ also depends on other parameters like $T$, $U_{\max}$, $\mathcal{F}$ and the initial policy $\boldsymbol{\pi}_1$. For simplicity, we omit those dependence unless necessary.

Given the above discussion, one natural question is that: **when is** $\Psi^\varepsilon$ **non-empty**, or equivalently, **when does a strategy** $\psi$ **with** $\varepsilon$**-steering gap exist**? In Sec. 4, we provide sufficient conditions and concrete examples to address this question in theory. Notably, we suggest conditions where $\Psi^\varepsilon$ is non-empty for any $\varepsilon > 0$, so that the condition $\Psi^{\varepsilon/|\mathcal{F}|} \neq \varnothing$ in Prop. 3.3 is realizable, even for large $|\mathcal{F}|$. After that, in Sec. 5, we introduce algorithms to solve our Obj. (1).

## 4 Existence of Steering Strategy with $\varepsilon$-Steering Gap

In this section, we identify sufficient conditions such that $\Psi^\varepsilon$ is non-empty. In Sec. 4.1, we start with the special case when $f^*$ is known, i.e. $\mathcal{F} = \{f^*\}$. The results will serve as basis when we study the general unknown model setting with $|\mathcal{F}| > 1$ in Sec. 4.2.

### 4.1 Existence when $f^*$ is Known: Natural Policy Gradient as an Example

In this section, we focus on a popular choice of learning dynamics called Natural Policy Gradient (NPG) dynamics (Kakade, 2001; Agarwal et al., 2021) (a.k.a. the replicator dynamics (Schuster and Sigmund, 1983)) with direct policy parameterization. NPG is a special case of the Policy Mirror Descent (PMD) (Xiao, 2022). For the readability, we stick to NPG in the main text, and in Appx. D.1, we formalize PMD and extend the results to the general PMD, which subsumes other learning dynamics, like the online gradient ascent (Zinkevich, 2003).

**Definition 4.1** (Natural Policy Gradient). For any $n \in [N], t \in [T], h \in [H], s_h \in \mathcal{S}$, the policy is updated by: $\pi_{t+1,h}^n(\cdot|s_h) \propto \pi_{t,h}^n(\cdot|s_h) \exp(\alpha \widehat{A}_{h|r^n+u_t^n}^{n,\boldsymbol{\pi}_t}(s_h,\cdot))$. Here $\widehat{A}_{h|r^n+u_t^n}^{n,\boldsymbol{\pi}_t}$ is some random estimation for the advantage value $A_{h|r^n+u_t^n}^{n,\boldsymbol{\pi}_t}$ with $\mathbb{E}_{\pi^n}[\widehat{A}_{h|r^n+u_t^n}^{n,\boldsymbol{\pi}_t}(s_h,\cdot)] = 0$.

We use $\widehat{A}_{|r+u}^{\boldsymbol{\pi}}$ (and $A_{|r+u}^{\boldsymbol{\pi}}$) to denote the concatenation of the values of all agents, horizon, states and actions. We only assume $\widehat{A}_{|r+u}^{\boldsymbol{\pi}_t}$ is controllable and has positive correlation with $A_{|r+u}^{\boldsymbol{\pi}_t}$ but could be biased, which we call the "general incentive driven" agents.

**Assumption B** (General Incentive Driven Agents).

$$\forall t \in [T], \quad \langle \mathbb{E}[\widehat{A}_{|r+u_t}^{\boldsymbol{\pi}_t}], A_{|r+u_t}^{\boldsymbol{\pi}_t} \rangle \geqslant \lambda_{\min} \|A_{|r+u_t}^{\boldsymbol{\pi}_t}\|_2^2, \quad \|\widehat{A}_{|r+u_t}^{\boldsymbol{\pi}_t}\|_2^2 \leqslant \lambda_{\max}^2 \|A_{|r+u_t}^{\boldsymbol{\pi}_t}\|_2^2,$$

Note that the exact NPG is captured by the special case with $\lambda_{\min} = \lambda_{\max} = 1$. For NPG, since the policy is always bounded away from 0. We will use $\Pi^+ := \{\boldsymbol{\pi}|\forall n, h, a_h, s_h : \pi_h^n(a_h|s_h) > 0\}$ to denote such feasible policy set. We state our main result below.

**Theorem 4.2** (Informal). *Suppose $\eta^{goal}$ is Lipschitz in $\boldsymbol{\pi}$, given any initial $\boldsymbol{\pi}_1 \in \Pi^+$, for any $\varepsilon > 0$, if the agents follow Def. 4.1 under Assump. B, and $T$ and $U_{\max}$ are large enough, we have $\Psi^\varepsilon \neq \varnothing$.*

Our result is strong in indicating the existence of a steering path for any feasible initialization. The proof is based on construction. The basic idea is to design the $\boldsymbol{u}_t$ so that $A_{|r+u_t}^{\boldsymbol{\pi}_t} \propto \log \frac{\boldsymbol{\pi}^*}{\boldsymbol{\pi}_t}$, for some target policy $\boldsymbol{\pi}^* \in \Pi^+$ (approximately) maximizing $\eta^{goal}$, then we can guarantee the convergence of $\boldsymbol{\pi}_t$ towards $\boldsymbol{\pi}^*$ under Assump. B. The main challenge here would be the design of $u_t$. We defer the details and the formal statements to Appx. D.

### 4.2 Existence when $f^*$ is Unknown: the Identifiable Model Class

Intuitively, when $f^*$ is unknown, if we can first use a few steering steps $\widetilde{T} < T$ to explore and identify $f^*$, and then steer the agents from $\boldsymbol{\pi}_{\widetilde{T}}$ to the desired policy within $T - \widetilde{T}$ steps given the identified $f^*$, we can expect $\Psi^\varepsilon \neq \varnothing$. Motivated by this insight, we introduce the following notion.

**Definition 4.3** (($\delta, T_{\mathcal{F}}^\delta$)-Identifiable). Given $\delta \in (0,1)$, we say $\mathcal{F}$ is ($\delta, T_{\mathcal{F}}^\delta$)-identifiable, if $\max_\psi \min_{f \in \mathcal{F}} \mathbb{E}_{\psi, f}[\mathbb{I}[f = f_{\mathrm{MLE}}]] \geqslant 1 - \delta$, where $\mathbb{I}[\mathcal{E}] = 1$ if $\mathcal{E}$ is true and otherwise 0; $f_{\mathrm{MLE}} := \arg\max_{f \in \mathcal{F}} \sum_{t=1}^{T_{\mathcal{F}}^\delta} \log f(\boldsymbol{\pi}_{t+1}|\boldsymbol{\pi}_t, \boldsymbol{u}_t)$.

Intuitively, $\mathcal{F}$ is ($\delta, T_{\mathcal{F}}^\delta$)-identifiable, if $\exists \psi$, s.t. after $T_{\mathcal{F}}^\delta$ steering steps, the hidden model $f$ can be identified by the Maximal Likelihood Estimation (MLE) with high probability. Next, we provide a general set of ($\delta, T_{\mathcal{F}}^\delta$)-identifiable function classes with $T_{\mathcal{F}}^\delta$ upper bounded for any $\delta \in (0,1)$.

**Example 4.4.** [One-Step Difference] If $\forall \boldsymbol{\pi} \in \Pi$, there exists a steering reward $\boldsymbol{u_\pi} \in \mathcal{U}$, s.t. $\min_{f, f' \in \mathcal{F}} \mathbb{H}^2(f(\cdot | \boldsymbol{\pi}, \boldsymbol{r} + \boldsymbol{u_\pi}), f'(\cdot | \boldsymbol{\pi}, \boldsymbol{r} + \boldsymbol{u_\pi})) \geqslant \zeta$, for some universal $\zeta > 0$, where $\mathbb{H}$ is the Hellinger distance, then for any $\delta \in (0, 1)$, $\mathcal{F}$ is $(\delta, T_{\mathcal{F}}^{\delta})$-identifiable with $T_{\mathcal{F}}^{\delta} = O(\zeta^{-1} \log(|\mathcal{F}|/\delta))$.

Based on Def. 4.3, we provide a sufficient condition when $\Psi^{\varepsilon}$ is non-empty.

**Theorem 4.5.** *[A Sufficient Condition for Existence] Given any* $\varepsilon > 0$, $\Psi_T^{\varepsilon}(\mathcal{F}; \boldsymbol{\pi}_1)^3 \neq \varnothing$, *if* $\exists \widetilde{T} < T$, *s.t., (1)* $\mathcal{F}$ *is* $(\frac{\varepsilon}{2\eta_{\max}}, \widetilde{T})$-*identifiable, (2)* $\Psi_{T-\widetilde{T}}^{\varepsilon/2}(\mathcal{F}; \boldsymbol{\pi}_{\widetilde{T}}) \neq \varnothing$ *for any possible* $\boldsymbol{\pi}_{\widetilde{T}}$ *generated at step* $\widetilde{T}$ *during the steering.*

We conclude this section by noting that, by Thm. 4.2, the above condition (2) is realistic for NPG (or more general PMD) dynamics. The proofs for all results in this section are deferred to Appx. E.

# 5 LEARNING (APPROXIMATELY) OPTIMAL STEERING STRATEGY

In this section, we investigate how to solve Obj. (1). Comparing with the episodic RL setting, the main challenge is to learn a history-dependent policy. Since the history space grows exponentially in $T$, directly solving Obj. (1) can be computationally intractable for large $T$. Therefore, the main focus of this section is to design tractable algorithms to overcome this challenge.

As a special case, when the model is known, i.e. $\mathcal{F} = \{f^*\}$, by the Markovian property, Obj. (1) reduces to a normal RL objective, and a state-dependent steering strategy $\psi : \Pi \rightarrow \mathcal{U}$ is already enough. For completeness, we include the algorithm but defer to Alg. 3 in Appx. B.4. In the rest of this section, we focus on the general case $|\mathcal{F}| > 1$. In Sec. 5.1, we investigate the solutions when $|\mathcal{F}|$ is small, and in Sec. 5.2, we study the more challenging case when $|\mathcal{F}|$ is large.

## 5.1 SMALL MODEL CLASS: DYNAMIC PROGRAMMING WITH MODEL BELIEF STATE

**A Partially Observable MDP Perspective** In fact, we can interpret Obj. (1) as learning the optimal policy in a POMDP, in which the hidden state is $(\boldsymbol{\pi}_t, f)$, i.e. a tuple containing the policy and the hidden model $f$ uniformly sampled from $\mathcal{F}$, and the mediator can only partially observe the policy $\boldsymbol{\pi}_t$. It is well-known that any POMDP can be lifted to the *belief MDP*, where the state is the *belief state* of the original POMDP. Then, the optimal policy in the belief MDP is exactly the optimal history-dependent policy in the original POMDP (Ibe, 2013). In our case, for each step $t \in [T]$, the belief state is $(\boldsymbol{\pi}_t, b_t)$, where $b_t := [\Pr(f | \{\boldsymbol{\pi}_{t'}, \boldsymbol{u}_{t'}\}_{t'=1}^t, \boldsymbol{\pi}_t)]_{f \in \mathcal{F}}$ is the "model belief state" defined to be the posterior distribution of models given the history of observations and actions. When $|\mathcal{F}|$ is small, the model belief state $b_t \in \mathcal{R}^{|\mathcal{F}|}$ is low dimensional and computable. Learning $\psi^*$ is tractable by running any RL algorithm on the lifted MDP. In Proc. 1, we show how to steer in this setting. We defer the detailed algorithm of learning such belief-state dependent strategy to Alg. 4 in Appx. B.5.

---

**Procedure 1:** The Steering Procedure when $|\mathcal{F}|$ is Small

---

1 **Input**: Model Set $\mathcal{F}$; Total step $T$;
2 Solving Obj. (1) by learning a belief state-dependent strategy $\psi_{\text{Belief}}^*$ by Alg. 4 with $\mathcal{F}$ and $T$.
3 Deploy $\psi_{\text{Belief}}^*$ to steer the real agents for $T$ steps.

---

## 5.2 LARGE MODEL CLASS: A FIRST-EXPLORE-THEN-EXPLOIT FRAMEWORK

When $|\mathcal{F}|$ is large, the method in Sec. 5.1 is inefficient since the belief state $b_t$ is high-dimensional. In fact, the above POMDP interpretation implies the intractability of Obj. (1) for large $|\mathcal{F}|$: the number of hidden states of the POMDP scales with $|\mathcal{F}|$. Therefore, instead of exactly solving Obj. (1), we turn to the First-Explore-Then-Exploit (FETE) framework as stated in Procedure 2.

The first $\widetilde{T} < T$ steps are the exploration phase, where we learn and deploy an exploration policy $\psi^{\text{Explore}}$ maximizing the probability of identifying the hidden model with the MLE estimator. The

---

[3]Here we highlight the dependence on initial policy, model, and time for clarity (see Footnote 2)

remaining $T - \widetilde{T}$ steps belong to the exploitation stage. We first estimate the true model by the MLE with the interaction history with real agents. Next, we learn an exploitation strategy to steer real agents for the rest $T - \widetilde{T}$ steps by solving Obj. (1) with $\mathcal{F} = \{f_{\text{MLE}}\}$, time $T - \widetilde{T}$ and the initial policy $\boldsymbol{\pi}_{\widetilde{T}+1}$, as if $f_{\text{MLE}}$ is the true model.

**Justification for** FETE   We cannot guarantee that Desiderata 1& 2 are achievable, because we do not exactly solve Obj. 1. However, if $\mathcal{F}$ is $(\delta/|\mathcal{F}|, T_{\mathcal{F}}^{\delta/|\mathcal{F}|})$-identifiable (Def. 4.3) and we choose $\widetilde{T} \geqslant T_{\mathcal{F}}^{\delta/|\mathcal{F}|}$, we can verify $\Pr(f_{\text{MLE}} = f^*) \geqslant 1 - \delta$ in FETE. Therefore, we can still expect the exploitation policy $\psi^{\text{Exploit}}$ steer the agents to approximately maximize $\eta^{\text{goal}}(\boldsymbol{\pi}_{T+1})$ with reasonable steering cost for the rest $T - \widetilde{T}$ steps.

---

**Procedure 2:** The Steering Procedure when $|\mathcal{F}|$ is Large (The FETE Framework)

---

1 **Input**: Model Set $\mathcal{F}$; Total step $T$; Exploration horizon $\widetilde{T}$;
2 /* —————————————————— Exploration Phase —————————————————— */
3 Learn an exploration strategy $\psi^{\text{Explore}} \leftarrow \arg\max_\psi \frac{1}{|\mathcal{F}|} \sum_{f \in \mathcal{F}} \mathbb{E}_{\boldsymbol{\pi}_1', \boldsymbol{u}_1', ..., \boldsymbol{\pi}_{\widetilde{T}+1}' \sim \psi, f} [\mathbb{I}[f = \arg\max_{f' \in \mathcal{F}} \sum_{t=1}^{\widetilde{T}} \log f'(\boldsymbol{\pi}_{t+1}' | \boldsymbol{\pi}_t', \boldsymbol{u}_t')]]$.
4 Deploy $\psi^{\text{Explore}}$ to steer the real agents and collect $\{\boldsymbol{\pi}_1, \boldsymbol{u}_1, ..., \boldsymbol{\pi}_{\widetilde{T}}, \boldsymbol{u}_{\widetilde{T}}, \boldsymbol{\pi}_{\widetilde{T}+1}\}$
5 /* —————————————————— Exploitation Phase —————————————————— */
6 Estimate $f_{\text{MLE}} \leftarrow \arg\max_{f \in \mathcal{F}} \sum_{t=1}^{\widetilde{T}} \log f(\boldsymbol{\pi}_{t+1} | \boldsymbol{\pi}_t, \boldsymbol{u}_t)$
7 Deploy $\psi^{\text{Exploit}} \leftarrow \arg\max_\psi \mathbb{E}_{\psi, f_{\text{MLE}}}[\beta \cdot \eta^{\text{goal}}(\boldsymbol{\pi}_{T-\widetilde{T}+1}) - \sum_{t=1}^{T-\widetilde{T}} \eta^{\text{cost}}(\boldsymbol{\pi}_t, \boldsymbol{u}_t) | \boldsymbol{\pi}_1 = \boldsymbol{\pi}_{\widetilde{T}+1}]$.

---

We conclude this section by highlighting the computational tractability of FETE. Note that when computing $\psi^{\text{Exploit}}$, we treat $f_{\text{MLE}}$ as the true model, so an history-independent $\psi^{\text{Exploit}}$ is enough. Therefore, the only part where we need to learn a history-dependent strategy is in the exploration stage, and the maximal history length is at most $\widetilde{T}$, which can be much smaller than $T$. Moreover, in some cases, it is already enough to just learn a history-independent $\psi^{\text{Explore}}$ to do the exploration (for example, the model class in Example 4.4).

**Comparison with Canyakmaz et al. (2024)**   Both SIAR-MPC in Canyakmaz et al. (2024) and our FETE (Procedure 2) adopt a first-explore-then-exploit structure. We examine the differences between two algorihms from two aspects: exploration strategy and model estimation strategy. Regarding exploration strategy, SIAR-MPC uses noise-based random exploration, whereas we adopt a more strategic approach, which uses the identification success rate as a signal to learn the exploration policy. Empirical results in Sec. 6.2 demonstrate the higher efficiency of our methods. In terms of model estimation strategy, SIAR-MPC estimates the hidden model by solving a constrained regression problem (Eq. (8) in Canyakmaz et al. (2024)), while we solve a MLE objective. In principle, our MLE objective is more general and can recover the regression problem in SIAR-MPC if choosing $\mathcal{F}$ to be the function class including Gaussian noise perturbed dynamics with side-information constraints introduced in Canyakmaz et al. (2024).

## 6 EXPERIMENTS

In this section, we discuss our experimental results. For more details of all experiments in this section (e.g. experiment setup and training details), we defer to Appx. G. The steering horizon is set to be $T = 500$, and all the error bar shows 95% confidence level. We denote $[x]^+ := \max\{0, x\}$.

### 6.1 LEARNING STEERING STRATEGIES WITH KNOWLEDGE OF $f^*$

**Normal-Form Stag Hunt Game**   In Fig. 1-(b), we compare the agents' dynamics with/without steering, where the agents learn to play the Stag Hunt Game in Fig. 1-(a). We report the experiment setup here. Both agents follow the exact NPG (Def. 4.1 with $\widehat{A}^{\boldsymbol{\pi}} = A^{\boldsymbol{\pi}}$) with fixed learning rate $\alpha = 0.01$. For the steering setup, we choose the total utility as $\eta^{\text{goal}}$, and use PPO to train the steering strategy (one can choose other RL or control algorithms besides PPO). We also conduct experiments in a representative zero-sum game 'Matching Pennies', which we defer the details to Appx. G.2.

**Grid World Stag Hunt Game: Learning Steering Strategy with Observations on Agents' Behaviors** In the previous experiments, we consider the direct parameterization and the state space $\mathcal{X} = \Pi \subset \mathbb{R}^4$ has low dimension. In real-world scenarios, the policy space $\Pi$ can be extremely rich and high-dimensional if the agents consider neural networks as policies. In addition, the mediator may not get access to the agents' exact policy $\pi$ because of privacy issues.

This motivates us to investigate the possibility of steering agents with observations on agents' behavior only (e.g. trajectories of agents in a game $G$), instead of the full observation of $\pi$. In Appx. F, we justify this setup and formalize it as a partially observable extension of our current framework. We consider the evaluation in a grid-world version of the Stag Hunt Game as shown in Fig. 2-(a). In this setting, the state space in game $G$ becomes pixel-based images, and both agents (blue and red) will adopt Convolutional Neural Networks (CNN) based policies with thousands of parameters and update with PPO. We train a steering strategy, which only takes the agents' recent trajectories as input to infer the steering reward. As shown in Fig. 2-(b), without direct usage of the agents' policy, we can still train a steering strategy towards desired solution.

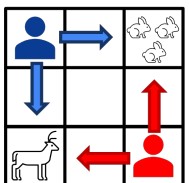 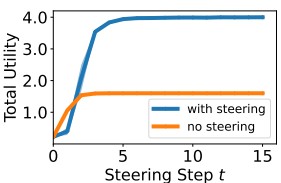

Figure 2: Grid-World Version of Stag Hunt Game. **Left**: Illustration of game. **Right**: The performance of agents with/without steering. Without steering, the agents converge to go for hares, which has sub-optimal utility. Under our learned steering strategy, the agents converge to a better equilibrium and chase the stag.

## 6.2 Learning Steering Strategies without Knowledge of $f^*$

**Small Model Set $|\mathcal{F}|$: Belief State Based Steering Strategy** In this part, we evaluate Proc. 1 designed for small $\mathcal{F}$. We consider the same normal-form Stag Hunt game and setup as Sec. 6.1, while the agents update by the NPG with a random learning rate $\alpha = [\xi]^+$, where $\xi \sim \mathcal{N}(\mu, 0.3^2)$. Here the mean value $\mu$ is unknown to the mediator, and we consider a model class $\mathcal{F} := \{f_{0.7}, f_{1.0}\}$ including two possible values of $\mu \in \{0.7, 1.0\}$. We report our experimental results in Table 1.

Table 1: **Evaluation for Proc. 1** (Averaged over 25 different initial $\pi_1$, see Appx. G.1).

(a) Performance in $f_{0.7}$

| | $p(\Delta_{\psi,T} \leqslant \varepsilon)$ | $C_{\psi,T}$ |
|---|---|---|
| $\psi^*_{0.7}$ | $0.99 \pm 0.01$ | $10.6 \pm 0.3$ |
| $\psi^*_{1.0}$ | $0.13 \pm 0.02$ | $7.6 \pm 0.2$ |
| $\psi^*_{\text{Belief}}$ | $0.87 \pm 0.05$ | $10.5 \pm 0.4$ |

(b) Performance in $f_{1.0}$

| | $p(\Delta_{\psi,T} \leqslant \varepsilon)$ | $C_{\psi,T}$ |
|---|---|---|
| $\psi^*_{0.7}$ | $1.00 \pm 0.00$ | $8.2 \pm 0.2$ |
| $\psi^*_{1.0}$ | $1.00 \pm 0.00$ | $5.6 \pm 0.2$ |
| $\psi^*_{\text{Belief}}$ | $0.99 \pm 0.01$ | $6.1 \pm 0.3$ |

Firstly, we demonstrate the suboptimal behavior if the mediator ignores the model uncertainty and just randomly deploys the optimal strategy of $f_{0.7}$ or $f_{1.0}$. To do this, we train the (history-independent) optimal steering strategy by Alg. 3, as if we know $f^* = f_{0.7}$ (or $f^* = f_{1.0}$), which we denote as $\psi^*_{0.7}$ (or $\psi^*_{1.0}$). To meet with our Desideratum 1, we first set the accuracy level $\varepsilon = 0.01$, and search the minimal $\beta$ so that the learned steering strategy can achieve $\varepsilon$-steering gap (see Appx. G.3.1). Because of the difference in $\mu$, we have $\beta = 70$ and $\beta = 20$ in training $\psi^*_{0.7}$ and $\psi^*_{1.0}$, respectively, and empirically, we observe that $\psi^*_{0.7}$ requires much larger steering reward than $\psi^*_{1.0}$. As we marked in red in Table 1-(a) and (b), because of the difference in the steering signal, $\psi^*_{0.7}$ consumes much higher steering cost to achieve the same accuracy level in $f_{1.0}$, and $\psi^*_{1.0}$ may fail to steer agents with $f_{0.7}$ to the desired accuracy. Next, we train another strategy $\psi^*_{\text{Belief}}$ via Alg. 4, which predicts the steering reward based on both the agents' policy $\pi$ and the belief state of the model. As we can see, $\psi^*_{\text{Belief}}$ can almost always achieve the desired $\varepsilon$-steering gap with reasonable steering cost.

**Large Model Set $|\mathcal{F}|$: The `FETE` Framework** In this part, we evaluate the `FETE` framework (Proc. 2 in Sec. 5.2). We consider an cooperative setting with $N = 10$ players. Each agent has two actions `A` and `B`, and the mediator only receives non-zero utility when all the agents cooperate together to take action `A`, i.e. $\eta^{\text{goal}}(\pi) := \prod_{n=1}^{N} \pi^n(\text{A})$. The agents do not have intrinsic rewards ($r = 0$), but the mediator's can steer them to maximize its own utility by providing additional steering rewards.

We consider "avaricious agents" with varying degrees of greediness, who tend to decrease the learning rates if the payments by mediator are high. Consequently, they require more steering steps to converge to the desired policies, potentially earning more incentive payments from the mediators. More concretely, the learning rate of agent $n$ is $\alpha_n := [\xi_n]^+$ with $\xi_n \sim \mathcal{N}(1.5 - \beta^n \cdot [V_{|\boldsymbol{u}}^{n,\boldsymbol{\pi}} - \lambda^n]^+, 0.5^2)$, where $\beta^n > 0$ is a scaling factor and $\lambda^n > 0$ is the threshold to exhibit

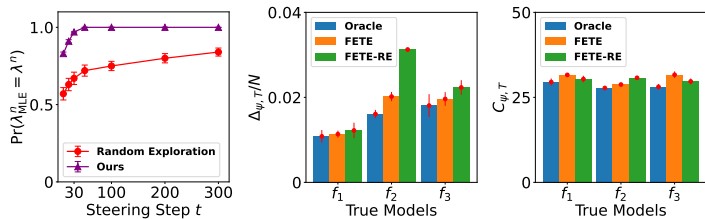

Figure 3: **Evaluation for Proc. 2. Left**: Accuracy of MLE estimator ($\lambda_{\mathrm{MLE}}^n$) after doing exploration for $t$ steps. Ours can achieve near 100% accuracy after 30 steering steps, while the random exploration takes more than 300 steps. **Middle and Right**: Average steering gap and steering cost of Oracle, FETE and FETE-RE. Our FETE achieves competitive performance comparing with Oracle, and significantly outperforms FETE-RE (adaption of SIAR-MPC (Canyakmaz et al., 2024)) to our setting) in terms of steering gap.

avaricious behavior. In our experiments, the model uncertainty comes from multiple possible realization of $\lambda^n \in \{0.25, 0.75, +\infty\}$, which results in an extremely large model class $\mathcal{F}$ with $|\mathcal{F}| = 3^{10}$. Here $\lambda^n = +\infty$ corresponds to normal agents whose learning rates are stable. The mediator does not know the agents' types $\{\lambda^n\}_{n \in [N]}$ in advance, and it can only observe one learning rate samples $\{\alpha_n\}_{n \in [N]}$ of agents per iteration $t \in [T]$ and estimate the true types from those samples. We consider the fixed initial policy with $\forall n \in [N]$, $\pi_1^n(\mathbb{A}) = 1 - \pi_1^n(\mathbb{B}) = 1/3$, and set the maximal steering reward $U_{\max} = 1.0$.

To understand the exploration challenge, note that, during the exploration phase, if the steering signal $\boldsymbol{u}$ is not strong enough, i.e. $V_{|\boldsymbol{u}}^{n,\boldsymbol{\pi}} < \lambda^n$, the mediator may fail to distinguish those avaricious agents from the normal ones, because they behave exactly the same. Such failure can lead to undesired outcomes in the exploitation phase: higher steering rewards can accelerate the convergence of normal agents, but can lead to larger steering gaps for avaricious agents.

We provide the evaluation results in Fig. 3. First, we compare the exploration efficiency. We can see the clear advantage of our strategic exploration in FETE (Procedure 2) compared with noise-based random exploration (Canyakmaz et al., 2024). Next, we compare the steering gaps and costs of three methods: (i) FETE; (ii) FETE-RE; (iii) Oracle – if the mediator knows $f^*$ in advance and solving Obj. (1) with $\mathcal{F} = \{f^*\}$. Here FETE-RE can be regarded as adaption of SIAR-MPC (Canyakmaz et al., 2024) to our case by replacing strategic exploration in FETE with random exploration (see Appx. G.4 for more explanation). We choose exploration horizon $\widetilde{T} = 30$ suggested by the previous exploration experiment, and report results for three realizations of $f^* \in \{f_1, f_2, f_3\}$. For $f_1$ and $f_2$, all the agents share $\lambda^n = 0.75$ and $+\infty$, respectively. $f_3$ is a mixed setup where $\lambda^n = 0.75$ for $1 \leqslant n \leqslant 5$ and $\lambda^n = +\infty$ for $5 < n \leqslant 10$. As we can see, comparing with Oracle, both the steering gap and cost of our FETE are competitive. Moreover, thanks to our strategic exploration method, FETE exhibits significant advantage over Canyakmaz et al. (2024) in terms of steering gaps.

## 7 CONCLUSION

In this paper, we introduce the problem of steering Markovian agents under model uncertainty. We provide theoretical foundations for this problem by formulating a novel optimization objective and providing existence results. Moreover, we design several algorithmic approaches suitable for varying degrees of model uncertainty in this problem class. We test their performances in different experimental settings and show their effectiveness. Our work opens up avenues for compelling open problems that merit future investigation. Firstly, future work could aim to identify superior optimization objectives that guarantee strictly better performances in terms of steering gap and cost than ours. Secondly, when applying our strategies in real-world applications, constraints on the steering reward budget could be added. Finally, the framework could be generalized to permit non-Markovian agents.

ACKNOWLEDGEMENT

The work is supported by ETH research grant and Swiss National Science Foundation (SNSF) Project Funding No. 200021-207343 and SNSF Starting Grant. VT is supported by an ETH AI Center Doctoral Fellowship. HHN is supported by the Swiss National Science Foundation under Eccellenza Grant 'Markets and Norms,' as well as under NCCR Automation Grant 51NF40-180545.

REPRODUCIBILITY STATEMENT

The code of all the experiments in this paper and the instructions for running can be found in `https://github.com/jiaweihhuang/Steering_Markovian_Agents`.

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

CONTENTS

## A    FREQUENTLY USED NOTATIONS

| Notation | Description |
|---|---|
| $G$ | A finite-horizon general-sum Markov Game |
| $N$ | The number of agents |
| $\mathcal{S}, \mathcal{A}$ | State space and action space of the game $G$ |
| $H$ | The horizon of the game $G$ |
| $\mathbb{P}$ | Transition function of the game $G$ |
| $r$ | Reward function of the game $G$ |
| $\boldsymbol{\pi}$ | The agents' policy (collection of policies of all agents) |
| $\boldsymbol{\pi}_1$ | The initial policy |
| $M$ | A finite-horizon Markov Decision Process (the steering MDP) |
| $\mathcal{X}, \mathcal{U}$ | State space and action space of $M$ |
| $T$ | The horizon of $M$ (i.e. the horizon of the steering dynamics) |
| $\mathbb{T}$ | (Stationary) Transition function of $M$ |
| $\eta^{\text{cost}}$ | The steering cost function of $M$ |
| $\psi$ | The history-dependent steering strategy by mediator |
| $u$ (or $u_t$ for a specific horizon $t$) | The steering reward function |
| $U_{\max}$ | The upper bound for steering reward |
| $f$ | Agents learning dynamics ($\mathbb{T} = f$ in the steering MDP) |
| $\eta^{\text{goal}}$ | The goal function of $M$ |
| $\mathcal{F}$ | The model class of agents dynamics (with finite candidates) |
| $\beta$ | Regularization coefficient in Obj. (1) |
| $C_{\psi,T}(f)$ | The total expected steering cost $\mathbb{E}_{\psi,f}\left[\sum_{t=1}^{T} \eta^{\text{cost}}(\boldsymbol{\pi}_t, \boldsymbol{u}_t)\right]$ |
| $\Delta_{\psi,T}(f)$ | The steering gap: $\mathbb{E}_{\psi,f}\left[\max_{\boldsymbol{\pi}} \eta^{\text{goal}}(\boldsymbol{\pi}) - \eta^{\text{goal}}(\boldsymbol{\pi}_{T+1})\right]$ |
| $\Psi$ | The collection of all history dependent policies |
| $\Psi^{\varepsilon}$ as a short note of $\Psi^{\varepsilon}_{T,U_{\max}}(\mathcal{F}; \boldsymbol{\pi}_1)$ | $\{\psi \in \Psi \mid \mathbb{E}_{\psi,f}[\max_{\boldsymbol{\pi}} \eta^{\text{goal}}(\boldsymbol{\pi}) - \eta^{\text{goal}}(\boldsymbol{\pi}_{T+1}) \mid \boldsymbol{\pi}_1] \leqslant \varepsilon\}$ |
| $Q^{n,\boldsymbol{\pi}}_{h\mid\boldsymbol{r}+\boldsymbol{u}}, V^{n,\boldsymbol{\pi}}_{h\mid\boldsymbol{r}+\boldsymbol{u}}, A^{n,\boldsymbol{\pi}}_{h\mid\boldsymbol{r}+\boldsymbol{u}}$ | The Q-value, V-value and advantage value functions for agent $n$ |
| $f_{\text{MLE}}$ | The Maximal Likelihood Estimator (introduced in Def. 4.3) |
| $b_t$ | Model belief state $[\Pr(f\mid\{\boldsymbol{\pi}_{t'}, u_{t'}\}_{t'=1}^{t}, \boldsymbol{\pi}_t)]_{f\in\mathcal{F}} \in \mathcal{R}^{\lvert\mathcal{F}\rvert}$ |
| $\psi^{\text{Explore}}/\psi^{\text{Exploit}}$ | The exploration/exploitation policy in FETE framework. |
| $O(\cdot), \Omega(\cdot), \Theta(\cdot), \widetilde{O}(\cdot), \widetilde{\Omega}(\cdot), \widetilde{\Theta}(\cdot)$ | Standard Big-O notations, $\widetilde{(\cdot)}$ omits the log terms. |

## B    MISSING DETAILS IN THE MAIN TEXT

### B.1    A REAL-WORLD SCENARIO THAT CAN BE MODELED AS A STAG HUNT GAME

As a real-world example, the innovation adaption can be modeled as a (multi-player) Stag Hunt game. Consider a situation involving a coordination problem where people can choose between an inferior/unsustainable communication or transportation technology that is cheap (the Gather action) and a superior technology that is sustainable but more expensive (the Hunt action). If more and more people buy products by the superior technology, the increasing profits can lead to the development of that technology and the decrease of price. Eventually, everyone can afford the price and benefit from the sustainable technology. In contrast, if people are trapped by the products of the inferior technology due to its low price, the long-run social welfare can be sub-optimal. The mediator's goal is to steer the population to adopt the superior technology.

### B.2    ADDITIONAL RELATED WORKS

We first complements the comparison with Zhang et al. (2023) in Sec. 1.1 by noting a minor but worth to be mentioned difference between our setting and (Zhang et al., 2023) in terms of incentive

schemes. While they consider that the mediator influences the agents' learning dynamics through a scalar payment function, we operate with additional steering rewards in a multi-dimensional reward vector space. As a result, there may not exist direct ways to translate the steering strategies between both settings, especially in the bandit feedback setting where only the sampled actions of agents can be observed (Zhang et al., 2023).

**Opponent Shaping**    In the RL literature a line of work focus on the problem of opponent shaping, where agents can influence each others learning by handing out rewards (Foerster et al., 2018; Yang et al., 2020; Willi et al., 2022; Lu et al., 2022; Willis et al., 2023; Zhao et al., 2022). Although the ways of influencing agents are similar to our setting, we study the problem of a mediator that acts outside the Markov Game and steers all the agents towards desired policies, while in opponent shaping the agents themselves learn to influence each other for their own interests.

**Learning Dynamics in Multi-Agent Systems**    In multi-agent setting, it is an important question to design learning dynamics and understand their convergence properties (Hernandez-Leal et al., 2017). Previous works has established near-optimal convergence guarantees to equilibra (Daskalakis et al., 2021; Cai et al., 2024). When the transition model of the multi-agent system is unknown, many previous works have studied how to conduct efficient exploration and learn equilibria under uncertainty (Jin et al., 2021; Bai et al., 2020; Zhang et al., 2021; Leonardos et al., 2021; Yardim et al., 2023; Huang et al., 2024b;a). However, most of these results only have guarantees on solving an arbitrary equilibrium when multiple equilibria exists, and it is unclear how to build algorithms based on them to reach some desired policies to maximize some goal functions.

**Mathematical Programming with Equilibrium Constraints (MPEC)**    MPEC generalises bilevel optimization to problems where the lower level consists of solving an equilibrium problem (Luo et al., 1996). (Li et al., 2020; Liu et al., 2022; Wang et al., 2022; 2023; Yang et al., 2022). These works consider variants of an MPEC and present gradient based approaches, most of which rely on computing hypergradients via the implicit function theorem and thus strong assumptions on the lower level problem, such as uniqueness of the equilibrium. Most games fail to satisfy such constraints. In contrast, our work makes no assumptions on the equilibrium structure and instead mild assumptions on the learning dynamics.

**Game Theory and Mechanism Design**    In Game Theory, a setup such as ours can be modelled as a Stackelberg game. Several works have considered finding Stackelberg equilibria using RL (Gerstgrasser and Parkes, 2023; Zhong et al., 2024) or gradient-based approaches (Fiez et al., 2020). Deng et al. (2019) showed how agents can manipulate learning algorithms to achieve more reward, as if they were playing a Stackelberg game. Related problems are implementation theory (Monderer and Tennenholtz, 2004) and equilibrium selection (Harsanyi and Selten, 1992). Moreover, the field of mechanism design has been concerned with creating economic games that implement certain outcomes as their equilibria. Several recent works have considered mechanism design on Markov Games (Curry et al., 2024; Baumann et al., 2020; Guo et al., 2023). In the case of congestion games, mechanisms have been proposed to circumvent the price of anarchy (Balcan et al., 2013; Paccagnan and Gairing, 2021; Roughgarden and Tardos, 2004), i.e. equililbria with low social welfare.

There is also a line of work has focused on control strategies for evolutionary games (Gong et al., 2022; Paarporn et al., 2018). However, the game and learning dynamics differ significantly from our setting. For a full survey of control-theoretic approaches, we refer the reader to Ratliff et al. (2019); Riehl et al. (2018).

**Bilevel Reinforcement Learning**    Bilevel RL considers the problem of designing an MDP—by for example changing the rewards—with a desireable optimal policy. Recently, several works have studied gradient-based approaches to find such good MDP configurations (Chen et al., 2022; Chakraborty et al., 2023; Shen et al., 2024; Thoma et al., 2024). While similar in some regards, in this setting we assume the lower level is a Markov Game instead of just an MDP. Moreover, our aim is not to design a game with a desireable equilibrium from scratch, but to take a given game and agent dynamics and steer them with minimal additional rewards to a desired outcome within a certain amount of time. Therefore our upper-level problem is a strategic decision-making problem, solved by RL instead of running gradient descent on some parameter space.

**Episodic RL and Non-Episodic RL**   Most of the existing RL literature focus on the episodic learning setup, where the entire interaction history can be divided into multiple episodes starting from the same initial state distribution(Dann and Brunskill, 2015; Dann et al., 2017). Comparing with this setting, our finite-horizon non-episodic setting is more challenging because the mediator cannot simply learn from repeated trial-and-error. Therefore, the learning criterions (e.g. no-regret (Azar et al., 2017; Jin et al., 2018) or sample complexity (Dann and Brunskill, 2015)) in episodic RL setting is not suitable in our case, which targets at finding a near-optimal policy in maximizing return. This motivates us to consider the new objective (Obj. (1)).

To our knowledge, most of the previous works use "non-episodic RL" to refer to the learning in infinite-horizon MDP. One popular setting is the infinite-horizon MDPs with stationary transitions, where people consider the discounted (Schulman et al., 2017; Dong et al., 2019) or average return (Auer et al., 2008; Wei et al., 2020). The infinit-horizon setting with non-stationary dynamics is known as the continual RL (Khetarpal et al., 2022; Abel et al., 2024), where the learners "never stops learning" and continue to adapt to the dynamics. Since we focus on the steering problem with *fixed and finite* horizon, the methodology in those works cannot be directly applied here.

Most importantly, we are also the first work to model the steering problem as a RL problem.

### B.3   A BRIEF INTRODUCTION TO MARKOV DECISION PROCESS

A finite-horizon Markov Decision Process is specified by a tuple $M := \{x_1, T, \mathcal{X}, \mathcal{U}, \mathbb{T}, (\eta, \eta^{\text{term}})\}$, where $x_1$ is the fixed initial state, $T$ is the horizon length, $\mathcal{X}$ is the state space, $\mathcal{U}$ is the action space. Besides, $\mathbb{T} := \{\mathbb{T}_t\}_{t \in [T]}$ with $\mathbb{T}_t : \mathcal{X} \times \mathcal{U} \to \Delta(\mathcal{X})$ denoting the transition function[4], $\eta := \{\eta_t\}_{t \in [T]}$ with $\eta_t : \mathcal{X} \times \mathcal{U} \to [0, 1]$ is the normal reward function and $\eta^{\text{term}} : \mathcal{X} \times \mathcal{U} \to [0, 1]$ denotes the additional terminal reward function. In this paper, without further specification, we will consider history dependent non-stationary policies $\Psi := \{\psi := \{\psi_1, ..., \psi_T\} | \forall t \in [T], \psi_t : (\mathcal{X} \times \mathcal{U})^{t-1} \times \mathcal{X} \to \Delta(\mathcal{U})\}$. Given a $\psi \in \Psi$, an episode of $M$ is generated by: $\forall t \in [T]$, $\boldsymbol{u}_t \sim \psi_t(\cdot | \{x_{t'}, \boldsymbol{u}_{t'}\}_{t'=1}^{t-1}, x_t)$, $\eta_t \leftarrow \eta_t(x_t, \boldsymbol{u}_t)$, $x_{t+1} \sim \mathbb{T}_t(\cdot | x_t, \boldsymbol{u}_t)$; $\eta^{\text{term}} \leftarrow \eta^{\text{term}}(x_{T+1})$;

### B.4   ALGORITHM FOR LEARNING OPTIMAL (HISTORY-INDEPENDENT) STRATEGY WHEN $f^*$ IS KNOWN

---
**Algorithm 3:** Learning with Known Steering Dynamics

---
1 **Input**: Model Set $\mathcal{F} := \{f^*\}$; Initial steering strategy $\psi^1 := \{\psi_t^1\}_{t \in [T]}$; Regularization coefficient $\beta$; Iteration number $K$;
2 **for** $k = 1, 2, ..., K$ **do**
3     Agents initialize with policy $\boldsymbol{\pi}_1^k$.
4     Sample trajectories with $\psi_{\zeta_k}$, $\forall t \in [T]$:

$$\boldsymbol{u}_t^k \sim \psi_t^k(\cdot | \boldsymbol{\pi}_t^k), \ \boldsymbol{\pi}_{t+1}^k \sim f^*(\cdot | \boldsymbol{\pi}_t^k, \boldsymbol{r} + \boldsymbol{u}_t^k), \ \eta_t^k = -\eta^{\text{cost}}(\boldsymbol{\pi}_t^k, \boldsymbol{u}_t^k).$$

5     Update $\psi^{k+1} \leftarrow \texttt{RLAlgorithm}(\psi^k, \{\boldsymbol{\pi}_t^k, \boldsymbol{u}_t^k, \eta_t^k\}_{t=1}^T \cup \{\beta \cdot \eta^{\text{goal}}(\boldsymbol{\pi}_{T+1}^k)\})$.
6 **end**
7 Output $\widehat{\psi}^* \leftarrow \psi_{\zeta_K}$.

---

### B.5   ALGORITHM FOR LEARNING BELIEF-STATE DEPENDENT STEERING STRATEGY

## C   MISSING PROOFS IN SECTION 3

**Proposition 3.3.** *[Justification for Obj. (1)] By solving Obj. (1): (1) $\psi^*$ is Pareto Optimal; (2) Given any $\varepsilon, \varepsilon' > 0$, if $\Psi^{\varepsilon/|\mathcal{F}|} \neq \varnothing$ and $\beta \geqslant \frac{U_{\max} N H T |\mathcal{F}|}{\varepsilon'}$, we have $\psi^* \in \Psi^{\varepsilon + \varepsilon'}$;*

---
[4]In this paper, we focus on stationary transition function, i.e. $\mathbb{T}_1 = ... = \mathbb{T}_T$.

---

**Algorithm 4:** Solving Obj. (1) by Learning Belief State-Dependent Strategy

---

1  **Input**: Model Set $\mathcal{F}$; Regularization coefficient $\beta$; Initial steering strategy $\psi^1 := \{\psi_t^1\}_{t=1}^T$;
   Iteration number $K$;

2  **for** $k = 1, 2, ..., K$ **do**

3     Sample $f \sim \text{Uniform}(\mathcal{F})$; Initialize $\boldsymbol{\pi}_1^k = \boldsymbol{\pi}_1$.

4     Sample trajectories with $\psi^k$ from simulator of $f$:

5         $\forall t \in [T] \quad b_t^k := \Pr(\cdot | \boldsymbol{\pi}_1^k, \boldsymbol{u}_1^k, ..., \boldsymbol{\pi}_{t-1}^k, \boldsymbol{u}_{t-1}^k, \boldsymbol{\pi}_t^k), \quad \boldsymbol{u}_t^k \sim \psi_t^k(\cdot | b_t^k, \boldsymbol{\pi}_t^k),$

6               $\boldsymbol{\pi}_{t+1}^k \sim f(\cdot | \boldsymbol{\pi}_t^k, \boldsymbol{r} + \boldsymbol{u}_t^k), \quad \eta_t^k \leftarrow -\eta^{\text{cost}}(\boldsymbol{\pi}_t^k, \boldsymbol{u}_t^k)$

7     Update $\psi^{k+1} \leftarrow \texttt{RLAlgorithm}(\psi^k, \{(\boldsymbol{\pi}_t^k, b_t^k), \boldsymbol{u}_t^k, \eta_t^k\}_{t=1}^T \cup \{\beta \cdot \eta^{\text{goal}}(\boldsymbol{\pi}_{T+1}^k)\}).$

8  **end**

9  **return** $\widehat{\psi}^* := \psi^K = \{\psi_t^K\}_{t=1}^T$

---

*Proof.* Suppose $\Psi^{\varepsilon/|\mathcal{F}|}$ is non-empty, we denote $\psi^{\varepsilon/|\mathcal{F}|}$ as one of the elements in $\Psi^{\varepsilon/|\mathcal{F}|}$. By definition, since $\max_{\boldsymbol{\pi}} \eta^{\text{goal}}(\boldsymbol{\pi})$ is fixed, we have:

$$\psi^* \leftarrow \arg\max_{\psi} \frac{1}{|\mathcal{F}|} \sum_{f \in \mathcal{F}} -\beta \Delta(\psi, f, T) - C(\psi, f, T).$$

If $\beta \geqslant \frac{U_{\max} NHT |\mathcal{F}|}{\varepsilon'}$, by definition,

$$0 \leqslant \Big( \frac{1}{|\mathcal{F}|} \sum_{f \in \mathcal{F}} -\beta \Delta(\psi^*, f, T) - C(\psi^*, f, T) \Big) - \Big( \frac{1}{|\mathcal{F}|} \sum_{f \in \mathcal{F}} -\beta \Delta(\psi^{\varepsilon/|\mathcal{F}|}, f, T) - C(\psi^{\varepsilon/|\mathcal{F}|}, f, T) \Big)$$

$$\leqslant \frac{1}{|\mathcal{F}|} \sum_{f \in \mathcal{F}} \beta \Big( \Delta(\psi^{\varepsilon/|\mathcal{F}|}, f, T) - \Delta(\psi^*, f, T) \Big) + U_{\max} NHT$$

$$\text{(the steering reward } u \in [0, U_{\max}])$$

$$\leqslant \frac{1}{|\mathcal{F}|} \sum_{f \in \mathcal{F}} \beta \Big( \frac{\varepsilon}{|\mathcal{F}|} - \Delta(\psi^*, f, T) \Big) + U_{\max} NHT \qquad (\psi^{\varepsilon/|\mathcal{F}|} \in \Psi_{T, U_{\max}}^{\varepsilon}(\mathcal{F}))$$

$$\leqslant \frac{U_{\max} NHT}{\varepsilon'} \Big( \frac{\varepsilon}{|\mathcal{F}|} - \frac{1}{|\mathcal{F}|} \sum_{f \in \mathcal{F}} \Delta(\psi^*, f, T) \Big) + U_{\max} NHT$$

As a direct observation, if $\mathbb{E}_{f \sim \text{Unif}(\mathcal{F})}[\Delta(\psi^*, f, T)] = \frac{1}{|\mathcal{F}|} \sum_{f \in \mathcal{F}} \Delta(\psi^*, f, T) > \frac{\varepsilon + \varepsilon'}{|\mathcal{F}|}$, the RHS will be strictly less than 0, which results in contradiction. Therefore, we must have

$$\forall f \in \mathcal{F}, \quad \Delta(\psi^*, f, T) \leqslant |\mathcal{F}| \cdot \mathbb{E}_{f \sim \text{Unif}(\mathcal{F})}[\Delta(\psi^*, f, T)] \leqslant \varepsilon + \varepsilon'.$$

which implies $\psi^* \in \Psi^{\varepsilon + \varepsilon'}$.

Next, we show the Pareto Optimality. If there exists $\psi$ and $f$ such that

- For all $f' \in \mathcal{F}$ with $f \neq f'$, $C(\psi^*, f, T) \geqslant C(\psi, f, T)$ and $\Delta(\psi^*, f, T) \geqslant \Delta(\psi, f, T)$;

- For $f$, either $C(\psi^*, f, T) > C(\psi, f, T)$ and $\Delta(\psi^*, f, T) \geqslant \Delta(\psi, f, T)$ or $C(\psi^*, f, T) \geqslant C(\psi, f, T)$ and $\Delta(\psi^*, f, T) > \Delta(\psi, f, T)$.

Therefore, we must have:

$$\frac{1}{|\mathcal{F}|} \sum_{f \in \mathcal{F}} \beta \Delta(\psi, f, T) - C(\psi, f, T) < \frac{1}{|\mathcal{F}|} \sum_{f \in \mathcal{F}} \beta \Delta(\psi^*, f, T) - C(\psi^*, f, T),$$

which conflicts with the optimality condition of Obj. (1). $\qquad\square$

**Remark C.1.** *Intuitively, Pareto optimality ensures the steering rewards predicted by $\psi^*$ are not necessarily large. For example, consider agents following NPG dynamics (Def. 4.1). Given any steering strategy $\psi$ and a constant $c > 0$, consider another steering strategy $\psi'(\cdot) := \psi(\cdot) + c$, i.e., by increasing the steering reward predicted by $\psi$ with $c$. Note that the learning dynamics under both $\psi$ and $\psi'$ are the same, and therefore the final steering gaps are the same. Pareto optimality implies that $\psi'$ will never be more preferable than $\psi$.*

# D    MISSING PROOFS FOR EXISTENCE WHEN THE TRUE MODEL $f^*$ IS KNOWN

In this section, we study the Policy Mirror Descent as a concrete example. In Appx. D.1, we provide more details about PMD. Then, we study the PMD with exact updates and stochastic updates in Appx. D.2.1 and D.2.2, respectively. The theorems in Sec. 4.1 will be subsumed as special cases.

## D.1    MORE DETAILS ABOUT POLICY MIRROR DESCENT

**Definition D.1** (Policy Mirror Descent). For each agent $n \in [N]$, the updates at step $t \in [T]$ follows:

$$\forall h \in [H], s_h \in \mathcal{S}, \quad \theta_{t+1,h}^n(\cdot|s_h) \leftarrow \theta_{t,h}^n(\cdot|s_h) + \alpha \widehat{A}_{h|r^n+u_t^n}^{n,\boldsymbol{\pi}_t}(s, \cdot), \quad \text{(Update in the mirror space)}$$

$$z_{t+1,h}^n(\cdot|s_h) \leftarrow (\nabla \phi^n)^{-1}(\theta_{t+1,h}^n(\cdot|s_h)) \quad \text{(Map } \theta \text{ back to the primal space)}$$

$$\pi_{t+1,h}^n(\cdot|s_h) \leftarrow \operatorname*{arg\,min}_{z \in \Delta(\mathcal{A}^n)} D_{\phi^n}(z, z_{t+1,h}^n(\cdot|s_h)), \quad \text{(Projection)}$$

Similar to Def. 4.1, here $\widehat{A}_{h|r^n+u_t^n}^{n,\boldsymbol{\pi}_t}$ is some random estimation for the advantage value $A_{h|r^n+u_t^n}^{n,\boldsymbol{\pi}_t}$ with $\mathbb{E}_{\pi^n}[\widehat{A}_{h|r^n+u_t^n}^{n,\boldsymbol{\pi}_t}(s_h, \cdot)] = 0$. Besides, $\theta_{t,h}^n \in \mathbb{R}^{|\mathcal{S}||\mathcal{A}|}$ denotes the variable in the dual space. $\phi^n : \operatorname{dom}(\phi^n) \to \mathbb{R}$ is a function satisfying Assump. C below, which gives the mirror map $\nabla \phi^n$; $(\nabla \phi^n)^{-1}$ is the inverse mirror map; $D_{\phi^n}(z, \widetilde{z}) := \phi^n(z) - \phi^n(\widetilde{z}) - \langle \nabla \phi^n(\widetilde{z}), z - \widetilde{z} \rangle$ is the Bregman divergence regarding $\phi^n$.

**Assumption C.** We assume for all $n \in [N]$, $\phi^n$ is $\mu$-strongly convex and essentially smooth, i.e. differentiable and $\|\nabla \phi^n(z^k)\| \to +\infty$ for any sequence $z^k \in \operatorname{dom}(\phi^n)$ converging to a point on the boundary of $\operatorname{dom}(\phi^n)$.

By Pythagorean Theorem and the strictly convexity of $D_{\phi^n}$, the projection $\boldsymbol{\pi}$ in Def. D.1 is unique.

**Lemma D.2.** *Given a convex set $\mathcal{C}$ and a function $\phi$ which is $\mu$-strongly convex on $\mathcal{C}$, we have*

$$\left\| \operatorname*{arg\,min}_{z \in \mathcal{C}} D_\phi(z, (\nabla \phi)^{-1}(\theta_1)) - \operatorname*{arg\,min}_{z \in \mathcal{C}} D_\phi(z, (\nabla \phi)^{-1}(\theta_2)) \right\| \leqslant \frac{1}{\mu} \|\theta_1 - \theta_2\|_2.$$

*Proof.* Given any dual variables $\theta_1$ and $\theta_2$, and their projection $z_1 := \operatorname*{arg\,min}_{z \in \mathcal{C}} D_\phi(z, (\nabla \phi)^{-1}(\theta_1))$ and $z_2 := \operatorname*{arg\,min}_{z \in \mathcal{C}} D_\phi(z, (\nabla \phi)^{-1}(\theta_2))$, by the first order optimality condition, we have:

$$\forall z \in \mathcal{C}, \quad \langle \nabla \phi(z_1) - \theta_1, z - z_1 \rangle \geqslant 0,$$
$$\langle \nabla \phi(z_2) - \theta_2, z - z_2 \rangle \geqslant 0$$

If we choose $z = z_2$ in the first equation and $z = z_1$ in the second equation, and sum together, we have:

$$\langle \theta_1 - \nabla \phi(z_1) + \nabla \phi(z_2) - \theta_2, z_1 - z_2 \rangle \geqslant 0,$$

By strongly convexity of $\phi$, the above implies:

$$\langle \theta_1 - \theta_2, z_1 - z_2 \rangle \geqslant \langle \nabla \phi(z_1) - \nabla \phi(z_2), z_1 - z_2 \rangle \geqslant \mu \cdot \|z_1 - z_2\|^2$$

Therefore,

$$\mu \|z_1 - z_2\| \leqslant \|\theta_1 - \theta_2\|,$$

and we finish the proof. $\square$

Next, we discuss some concrete examples.

**Example D.3** (Natural Policy Gradient). If we consider the mirror map and Bregman Divergence generated by $\phi^n(z) := \sum_{a^n \in \mathcal{A}^n} z(a^n) \log z(a^n)$, we have $D_\phi^n(z_1, z_2) = \mathrm{KL}(z_1 \| z_2)$, and recover the NPG in Def. 4.1. Note that $\phi^n$ is 1-strongly convex on the convex set $\Delta(\mathcal{A}^n)$, Assump. C is satisfied with $\mu = 1$.

**Example D.4** (Online Gradient Ascent (Zinkevich, 2003)). If we consider the Euclidean distance generated by $l_2$-norm $\phi^n(z) = \frac{1}{2}\|z\|_2^2$, we recover the projected gradient ascent

**Definition D.5.** For each agent $n \in [N]$, the updates at step $t \in [T]$ follows:

$$\forall h \in [H], s_h \in \mathcal{S}, \quad \pi_{t+1,h}^n(\cdot|s_h) \leftarrow \text{Proj}_{\Delta(\mathcal{A}^n)}(\pi_{t,h}^n(\cdot|s_h) + \alpha \widehat{A}_{h|r^n+u_t^n}^{n,\boldsymbol{\pi}_t}(s,\cdot)),$$

Note that the projection with Euclidean distance is 1-Lipschitz, Assump. C is satisfied with $\mu = 1$.

**Other Notations and Remarks** In the following, we use $\Pi^+$ to be the "feasible policy set" (for NPG in Def. 4.1, $\Pi^+$ refers to be set of policies bounded away from 0), such that for any $\boldsymbol{\pi} \in \Pi^+$, there exists a dual variable $\theta$ corresponding to $\boldsymbol{\pi}$, i.e.,

$$\forall n \in [N], h \in [H], s_h \in \mathcal{S}, \quad \pi_h^n(\cdot|s_h) \leftarrow \underset{z \in \Delta(\mathcal{A}^n)}{\arg\min} D_{\phi^n}(z, (\nabla\phi^n)^{-1}(\theta_h^n(\cdot|s_h))).$$

In the following Lem. D.6, we show that constant shift in $\theta_{t,h}^n(\cdot|s_h)$ does not change the projection result. Therefore, when we say the dual variable $\theta$ associated with a given policy $\boldsymbol{\pi}$, we only consider those $\theta$ satisfying $\mathbb{E}_{a_h^n \sim \pi_h^n}[\theta_h^n(a_h^n|s_h)] = 0$.

**Lemma D.6** (Constant Shift does not Change the Projection). *For any $n \in [N]$, regularizer $\phi^n$ satisfying conditions in Assump. C, and any $\theta \in \mathbb{R}^{|\mathcal{A}^n|}$, consider the constant vector $c\boldsymbol{1}$, where $c \in \mathbb{R}$ is a constant and $\boldsymbol{1} = \{1, 1, ..., 1\} \in \mathbb{R}^{|\mathcal{A}^n|}$, we have:*

$$\underset{z \in \Delta(\mathcal{A}^n)}{\arg\min} D_{\phi^n}(z, (\nabla\phi^n)^{-1}(\theta)) = \underset{z \in \Delta(\mathcal{A}^n)}{\arg\min} D_{\phi^n}(z, (\nabla\phi^n)^{-1}(\theta + c\boldsymbol{1}))$$

*Proof.*

$$\begin{aligned}
&\arg \min_{z \in \Delta(\mathcal{A}^n)} D_{\phi^n}(z, (\nabla\phi^n)^{-1}(\theta + c\boldsymbol{1})) \\
&= \arg \min_{z \in \Delta(\mathcal{A}^n)} \phi^n(z) - \langle \theta + c\boldsymbol{1}, z \rangle \\
&= \arg \min_{z \in \Delta(\mathcal{A}^n)} \phi^n(z) - \langle \theta, z \rangle + c \qquad\qquad \text{(we have constraints that } z \in \Delta(\mathcal{A}^n)) \\
&= \arg \min_{z \in \Delta(\mathcal{A}^n)} \phi^n(z) - \langle \theta, z \rangle \\
&= \arg \min_{z \in \Delta(\mathcal{A}^n)} D_{\phi^n}(z, \theta).
\end{aligned}$$

$\square$

## D.2 PROOFS FOR THE EXISTENCE OF DESIRED STEERING STRATEGY

We first formally introduce the Lipschitz condition that Thm. 4.2 requires.

**Assumption D** ($\eta^{\text{goal}}$ is $L$-Lipschitz). For any $\boldsymbol{\pi}, \boldsymbol{\pi}' \in \Pi$, $|\eta^{\text{goal}}(\boldsymbol{\pi}) - \eta^{\text{goal}}(\boldsymbol{\pi}')| \leqslant L\|\boldsymbol{\pi} - \boldsymbol{\pi}'\|_2$.

In the following, in Appx. D.2.1, as a warm-up, we start with the exact case when the estimation $\widehat{A}^{\boldsymbol{\pi}}$ is exactly the true advantage value $A^{\boldsymbol{\pi}}$ (which can be regarded as a special case of Assump. B). Then, in Appx. D.2.2, we study the general setting and prove Thm. 4.2 as a special case of PMD.

### D.2.1 SPECIAL CASE: PMD WITH EXACT ADVANTAGE-VALUE

**Lemma D.7** (Existence of Steering Path between Feasible Policies). *Consider two feasible policies $\boldsymbol{\pi}, \widetilde{\boldsymbol{\pi}}$ which are induced by dual variables $\{\theta_{1,h}^n\}_{h \in [H], n \in [N]}$ and $\{\widetilde{\theta}_h^n\}_{h \in [H], n \in [N]}$, respectively. If the agents follow Def. D.1 with exact $Q$ value and start with $\boldsymbol{\pi}_1 = \boldsymbol{\pi}$, as long as $U_{\max} \geqslant 2H + \frac{2}{\alpha T}(\max_{n,h,s_h,a_h^n} |\widetilde{\theta}_h^n(a_h^n|s_h) - \theta_h^n(a_h^n|s_h) - \mathbb{E}_{\bar{a}_h^n \sim \pi_{t,h}^n(\cdot|s_h)}[\widetilde{\theta}_h^n(\bar{a}_h^n|s_h) - \theta_h^n(\widetilde{a}_h^n|s_h)]|)$, there exists a (history-independent) steering strategy $\psi := \{\psi_t\}_{t \in [T]}$ with $\psi_t : \Pi^+ \to \mathcal{U}$, s.t., $\boldsymbol{\pi}_{T+1} = \widetilde{\boldsymbol{\pi}}$.*

*Proof.* For agent $n \in [N]$, given a $\boldsymbol{\pi}_t$, we consider the following steering reward functions

$$u_{t,h}^n(s_h, a_h^n) = \nu_{t,h}^n(s_h, a_h^n) - A_{h|r^n}^{n,\boldsymbol{\pi}_t}(s_h, a_h^n) - \mathbb{E}_{\widetilde{a}^n \sim \pi_{t,h}^n(\cdot|s_h)}[\nu_{t,h}^n(s_h, \widetilde{a}_h^n) - A_{h|r^n}^{n,\boldsymbol{\pi}_t}(s_h, \widetilde{a}_h^n)]$$

$$- \min_{\bar{s}_h, \bar{a}_h^n} \{ \nu_{t,h}^n(\bar{s}_h, \bar{a}_h^n) - A_{h|r^n}^{n,\boldsymbol{\pi}_t}(\bar{s}_h, \bar{a}_h^n) - \mathbb{E}_{\widetilde{a}^n \sim \pi_{t,h}^n(\cdot|s_h)}[\nu_{t,h}^n(\bar{s}_h, \widetilde{a}_h^n) - A_{h|r^n}^{n,\boldsymbol{\pi}_t}(\bar{s}_h, \widetilde{a}_h^n)] \},$$

where $\nu_{t,h}^n : \mathcal{S} \times \mathcal{A}^n \to \mathbb{R}$ will be defined later. By construction, we have:

$$\mathbb{E}_{a_h^n \sim \pi_{t,h}^n(\cdot|s_h)}[u_{t,h}^n(s_h, a_h^n)] \tag{2}$$

$$= - \min_{\bar{s}_h, \bar{a}_h^n} \{ \nu_{t,h}^n(\bar{s}_h, \bar{a}_h^n) - A_{h|r^n}^{n,\boldsymbol{\pi}_t}(\bar{s}_h, \bar{a}_h^n) - \mathbb{E}_{\widetilde{a}^n \sim \pi_{t,h}^n(\cdot|s_h)}[\nu_{t,h}^n(\bar{s}_h, \widetilde{a}_h^n) - A_{h|r^n}^{n,\boldsymbol{\pi}_t}(\bar{s}_h, \widetilde{a}_h^n)] \}, \tag{3}$$

which is a constant and independent w.r.t. $s_h, a_h^n$. Besides, by definition, we can ensure the non-negativity of $u_{t,h}^n$. As a result,

$$\forall t \in [T], \quad Q_{h|r^n + u_t^n}^{t,\boldsymbol{\pi}_t}(s_h, a_h^n)$$
$$= A_{h|r^n}^{t,\boldsymbol{\pi}_t}(s_h, a_h^n) + u_{t,h}^n(s_h, a_h^n) + C_h(s_h) \tag{Eq. (3)}$$
$$= \nu_{t,h}^n(s_h, a_h^n) + C'_h(s_h). \tag{4}$$

where we use $C_h(s_h)$ and $C'_h(s_h)$ to denote some state-dependent but action-independent value. According to Lem. D.6, under the above steering reward design, the dynamics of $\boldsymbol{\pi}_1, ..., \boldsymbol{\pi}_t, ..., \boldsymbol{\pi}_T$ can be described by the following dynamics:

$$\forall t \in [T], \ \forall n \in [N], h \in [H], s_h \in \mathcal{S} : \quad \theta_{t+1,h}^n(\cdot|s_h) \leftarrow \theta_{t,h}^n(\cdot|s_h) + \alpha \nu_{t,h}^n(s_h, a_h^n) \tag{5}$$

$$\pi_{t+1,h}^n(\cdot|s_h) \leftarrow \arg\min_{z \in \Delta(\mathcal{A}^n)} D_{\phi^n}(z, \theta_{t+1,h}^n(\cdot|s_h)), \tag{6}$$

Now we consider the following choice of $\nu_{t,h}^n$:

$$\nu_{t,h}^n(s_h, a_h^n) = \frac{\widetilde{\theta}_h^n(a_h^n|s_h) - \theta_h^n(a_h^n|s_h)}{\alpha T},$$

which implies $\theta_{T+1} = \widetilde{\theta}$, and therefore, $\boldsymbol{\pi}_{T+1} = \widetilde{\boldsymbol{\pi}}$. Besides, the steering reward function can be upper bounded by:

$$u_{t,h}^n(s_h, a_h^n) \leqslant 2 \max_{\bar{s}_h, \bar{a}_h^n} |\nu_{t,h}^n(\bar{s}_h, \bar{a}_h^n) - A_{h|r^n}^{n,\boldsymbol{\pi}_t}(\bar{s}_h, \bar{a}_h^n) - \mathbb{E}_{\widetilde{a}^n \sim \pi_{t,h}^n(\cdot|s_h)}[\nu_{t,h}^n(\bar{s}_h, \widetilde{a}_h^n) - A_{h|r^n}^{n,\boldsymbol{\pi}_t}(\bar{s}_h, \widetilde{a}_h^n)]|$$

$$\leqslant 2H + \frac{2}{\alpha T}( \max_{n,h,s_h,a_h^n} |\widetilde{\theta}_h^n(a_h^n|s_h) - \theta_h^n(a_h^n|s_h)|),$$

which implies the appropriate choice of $U_{\max}$.

$$\square$$

**Theorem D.8.** *Under Assump. D, given the initial $\boldsymbol{\pi}_1 := \boldsymbol{\pi} \in \Pi^+$, for any $T \geqslant 1$ and $\varepsilon > 0$, if the agents follow Def. 4.1 with exact Q value, then $\Psi_{T,U_{\max}}^\varepsilon \neq \varnothing$ if the following conditions are satisfied:*

- *There exists feasible $\widetilde{\boldsymbol{\pi}} \in \Pi^+$ such that $\eta^{goal}(\widetilde{\boldsymbol{\pi}}) \geqslant \max_{\boldsymbol{\pi}} \eta^{goal}(\boldsymbol{\pi}) - \varepsilon$*

- *Denote $\theta$ and $\widetilde{\theta}$ as the dual variables associated with $\boldsymbol{\pi}$ and $\widetilde{\boldsymbol{\pi}}$, respectively. We require $U_{\max} \geqslant 2H + \frac{2}{\alpha T}(\max_{n,h,s_h,a_h^n} |\widetilde{\theta}_h^n(a_h^n|s_h) - \theta_h^n(a_h^n|s_h)|)$*

*Proof.* The proof is a directly application of Lem. D.7. $\square$

**NPG as a Special Case** For NPG, we have the following results.

**Lemma D.9.** *Given $\forall \boldsymbol{\pi}, \widetilde{\boldsymbol{\pi}} \in \Pi^+$, $T \geqslant 1$, if the agents follow Def. 4.1 with exact adv-value and start from $\boldsymbol{\pi}_1 = \boldsymbol{\pi}$, by choosing $U_{\max}$ appropriately, there exists a (history-independent) steering strategy $\psi := \{\psi_t\}_{t \in [T]}$ with $\psi_t : \Pi^+ \to \mathcal{U}$, s.t., $\boldsymbol{\pi}_{T+1} = \widetilde{\boldsymbol{\pi}}$.*

**Theorem D.10.** *Under Assump. D, given any initial $\boldsymbol{\pi}_1 \in \Pi^+$, for any $T \geqslant 1$ and $\varepsilon > 0$, if the agents follow Def. 4.1 with exact Q value, by choosing $U_{\max}$ appropriately, we have $\Psi^\varepsilon \neq \varnothing$.*

**Proof for Lem. D.9 and Thm. D.10** The proof is by directly applying Lem. D.7 and Thm. D.8 since NPG is a special case of PMD with KL-Divergence as Bregman Divergence. For any $\boldsymbol{\pi}, \widetilde{\boldsymbol{\pi}} \in \Pi^+$, we consider the dual variables $\theta, \widetilde{\theta}$ such that:

$$\theta_h^n(\cdot|s_h) = \log \pi_h^n(\cdot|s_h) - \mathbb{E}_{a_h^n \sim \pi_h^n}[\log \pi_h^n(a_h^n|s_h)], \quad \widetilde{\theta}_h^n(\cdot|s_h) = \log \widetilde{\pi}_h^n(\cdot|s_h) - \mathbb{E}_{a_h^n \sim \pi_h^n}[\log \widetilde{\pi}_h^n(a_h^n|s_h)]. \tag{7}$$

**Choice of $U_{\max}$ in Lem. D.9** By applying Lem. D.7 and Thm. D.8, we consider the following choice of $U_{\max}$

$$U_{\max} \geqslant 2H + \frac{2}{\alpha T}\big(\max_{n,h,s_h,a_h} |\log \frac{\widetilde{\pi}_h^n(s_h, a_h^n)}{\pi_h^n(s_h, a_h^n)} - \mathbb{E}_{\widetilde{a}_h^n \sim \pi_h^n}[\log \frac{\widetilde{\pi}_h^n(s_h, \widetilde{a}_h^n)}{\pi_h^n(s_h, \widetilde{a}_h^n)}]|\big). \tag{8}$$

**Choice of $U_{\max}$ in Thm. D.10** We denote $\boldsymbol{\pi}^* \in \arg\max_{\boldsymbol{\pi} \in \Pi} \eta^{\text{goal}}(\boldsymbol{\pi}) \notin \Pi^+$.

When $\boldsymbol{\pi}^* \in \Pi^+$, we can directly apply Thm. D.8 with $\widetilde{\boldsymbol{\pi}} \leftarrow \boldsymbol{\pi}^*$, and choosing $U_{\max}$ correspondingly following Eq. (8).

However, in some cases, $\boldsymbol{\pi}^* \notin \Pi^+$ because it takes deterministic action in some states. In that case, since $\eta^{\text{goal}}$ is $L$-Lipschitz in $\boldsymbol{\pi}$, we can consider the mixture policy $\widetilde{\boldsymbol{\pi}} := (1 - O(\frac{\varepsilon}{L}))\boldsymbol{\pi}^* + O(\frac{\varepsilon}{L})\boldsymbol{\pi}_{\text{Uniform}}$, where $\boldsymbol{\pi}_{\text{Uniform}}$ is the uniform policy. As a result, we have $\widetilde{\boldsymbol{\pi}} \in \Pi^+$ as well as $\eta^{\text{goal}}(\widetilde{\boldsymbol{\pi}}) \geqslant \max_{\boldsymbol{\pi} \in \Pi} \eta^{\text{goal}}(\boldsymbol{\pi}) - \varepsilon$. Then the $U_{\max}$ can be chosen following Eq. (8).

### D.2.2 THE GENERAL INCENTIVE DRIVEN AGENTS UNDER ASSUMP. B

**Theorem D.11** (Formal Version of Thm. 4.2 for the general PMD)**.** *Under Assump. D and Assump. C, given the initial $\boldsymbol{\pi}_1 := \boldsymbol{\pi} \in \Pi^+$, for any $\varepsilon > 0$, if the agents follow Def. 4.1 under the Assump. B, then $\Psi_{T,U_{\max}}^\varepsilon \neq \varnothing$ if the following conditions are satisfied:*

- *There exists feasible $\widetilde{\boldsymbol{\pi}} \in \Pi^+$ such that $\eta^{goal}(\widetilde{\boldsymbol{\pi}}) \geqslant \max_{\boldsymbol{\pi}} \eta^{goal}(\boldsymbol{\pi}) - \frac{\varepsilon}{2}$*

- *Denote $\theta$ and $\widetilde{\theta}$ as the dual variables associated with $\boldsymbol{\pi}$ and $\widetilde{\boldsymbol{\pi}}$, respectively. We require $U_{\max} \geqslant 2(H + \frac{\lambda_{\min}}{\alpha\lambda_{\max}^2}(1 + \frac{\lambda_{\min}}{\lambda_{\max}})^T\|\widetilde{\theta} - \theta\|_2)$ and $T = \Theta(\frac{\lambda_{\max}^2}{\lambda_{\min}^2} \log \frac{L\|\widetilde{\theta}-\theta\|_2}{\mu\varepsilon})$.*

**Remark D.12.** *In Thm. D.2.2, our bound for $U_{\max}$ here is just a worst-case bound to handle the noisy updates in the worst case. With high probability, the dual variable $\theta_t$ will converge to $\widetilde{\theta}$ and the steering reward does not have to be as large as $U_{\max}$.*

*Proof.* Given a $\boldsymbol{\pi}_t$, we consider the following steering reward $u_t$:

$$u_{t,h}^n(s_h, a_h^n) = \nu_{t,h}^n(s_h, a_h^n, \boldsymbol{\pi}_t) - A_{h|r^n}^{n,\boldsymbol{\pi}_t}(s_h, a_h^n) - \mathbb{E}_{\widetilde{a}^n \sim \pi_{t,h}^n(\cdot|s_h)}[\nu_{t,h}^n(s_h, \widetilde{a}_h^n, \boldsymbol{\pi}_t) - A_{h|r^n}^{n,\boldsymbol{\pi}_t}(s_h, \widetilde{a}_h^n)]$$
$$- \min_{\bar{s}_h, \bar{a}_h^n}\{\nu_{t,h}^n(\bar{s}_h, \bar{a}_h^n, \boldsymbol{\pi}_t) - A_{h|r^n}^{n,\boldsymbol{\pi}_t}(\bar{s}_h, \bar{a}_h^n) - \mathbb{E}_{\widetilde{a}^n \sim \pi_{t,h}^n(\cdot|s_h)}[\nu_{t,h}^n(\bar{s}_h, \widetilde{a}_h^n, \boldsymbol{\pi}_t) - A_{h|r^n}^{n,\boldsymbol{\pi}_t}(\bar{s}_h, \widetilde{a}_h^n)]\},$$

Here we choose $\nu_{t,h}^n(s_h, a_h^n, \boldsymbol{\pi}_t) := \frac{1}{\gamma} \cdot (\widetilde{\theta}_h^n(a_h^n|s_h) - \theta_{t+1,h}^n(a_h^n|s_h))$, where $\widetilde{\theta}$ denotes the dual variable of policy $\widetilde{\boldsymbol{\pi}}$ and $\gamma$ will be determined later. Comparing with the design in the proof of Thm. D.8, here the "driven term" $\nu_h^n$ need to depend on $\boldsymbol{\pi}_t$ because of the randomness in updates.

As we can see, $u_{t,h}^n(s_h, a_h^n) \geqslant 0$, and for each $t$, we have:

$$\mathbb{E}[\|\widetilde{\theta} - \theta_{t+1}\|_2^2] = \mathbb{E}[\|\widetilde{\theta} - \theta_t\|_2^2] - 2\mathbb{E}[\langle \widetilde{\theta} - \theta_t, \theta_{t+1} - \theta_t \rangle] + \mathbb{E}[\|\theta_{t+1} - \theta_t\|_2^2]$$
$$= \mathbb{E}[\|\widetilde{\theta} - \theta_t\|_2^2] - 2\alpha\mathbb{E}[\langle \widetilde{\theta} - \theta_t, \widehat{A}_{|\boldsymbol{r}+\boldsymbol{u}_t}^{\boldsymbol{\pi}_t} \rangle] + \mathbb{E}[\|\widehat{A}_{|\boldsymbol{r}+\boldsymbol{u}_t}^{\boldsymbol{\pi}_t}\|_2^2]$$
$$\leqslant (1 - 2\lambda_{\min}\frac{\alpha}{\gamma} + \lambda_{\max}^2\frac{\alpha^2}{\gamma^2})\mathbb{E}[\|\widetilde{\theta} - \theta_t\|_2^2],$$

which implies

$$\mathbb{E}[\|\widetilde{\theta} - \theta_{T+1}\|_2^2] \leqslant (1 - 2\lambda_{\min}\frac{\alpha}{\gamma} + \lambda_{\max}^2\frac{\alpha^2}{\gamma^2})^T\|\widetilde{\theta} - \theta\|_2^2.$$

We consider the choice $\gamma = \frac{\lambda_{\max}^2 \alpha}{\lambda_{\min}}$, which implies,

$$\mathbb{E}[\|\widetilde{\theta} - \theta_{T+1}\|_2^2] \leqslant (1 - \frac{\lambda_{\min}^2}{\lambda_{\max}^2})^T \|\widetilde{\theta} - \theta\|_2^2.$$

When $T = 2c_0 \frac{\lambda_{\max}^2}{\lambda_{\min}^2} \log \frac{2L\|\widetilde{\theta}-\theta\|_2}{\mu\varepsilon} \geqslant c_0 \log_{1-\frac{\lambda_{\min}^2}{\lambda_{\max}^2}} \left( \frac{\nu^2 \varepsilon^2}{2L^2 \|\widetilde{\theta}-\theta\|_2^2} \right)$ for some constant $c_0$, we have:

$$\mathbb{E}[\|\widetilde{\theta} - \theta_{T+1}\|_2] \leqslant \frac{\mu\varepsilon}{2L},$$

which implies,

$$\mathbb{E}[\eta(\boldsymbol{\pi}^*) - \eta^{\text{goal}}(\boldsymbol{\pi}_{T+1})] \leqslant \frac{\varepsilon}{2} + L\mathbb{E}[\|\widetilde{\boldsymbol{\pi}} - \boldsymbol{\pi}_{T+1}\|_2] \leqslant \frac{\varepsilon}{2} + \frac{L}{\mu}\mathbb{E}[\|\widetilde{\theta} - \theta_{T+1}\|_2] = \varepsilon.$$

Next, we discuss the choice of $U_{\max}$, by Assump. B, we know,

$$\|\widetilde{\theta} - \theta_{t+1}\|_2 = \|\widetilde{\theta} - \theta_t - \alpha \widehat{A}_{|\boldsymbol{r}+\boldsymbol{u}_t}^{\boldsymbol{\pi}_t}\|_2 \leqslant \|\widetilde{\theta} - \theta_t\|_2 + \alpha \|\widehat{A}_{|\boldsymbol{r}+\boldsymbol{u}_t}^{\boldsymbol{\pi}_t}\|_2$$

$$\leqslant \|\widetilde{\theta} - \theta_t\|_2 + \alpha \lambda_{\max} \|A_{|\boldsymbol{r}+\boldsymbol{u}_t}^{\boldsymbol{\pi}_t}\|_2$$

$$\leqslant (1 + \frac{\lambda_{\min}}{\lambda_{\max}}) \|\widetilde{\theta} - \theta_t\|_2$$

where we use the fact that $\|A_{|\boldsymbol{r}+\boldsymbol{u}_\tau}^{\boldsymbol{\pi}_\tau}\|_2 = \frac{1}{\gamma}\|\widetilde{\theta} - \theta_\tau\|_2$ and our choice of $\gamma$. Therefore, for all $t \in [T]$, $\|\widetilde{\theta} - \theta_t\|_2 \leqslant (1 + \frac{\lambda_{\min}}{\lambda_{\max}})^T \|\widetilde{\theta} - \theta\|_2$. To ensure our design of $u_{t,h}^n$ is feasible, we need to set:

$$U_{\max} = 2(H + \frac{1}{\gamma}(1 + \frac{\lambda_{\min}}{\lambda_{\max}})^T \|\widetilde{\theta} - \theta\|_2)$$

$$= 2(H + \frac{\lambda_{\min}}{\alpha \lambda_{\max}^2}(1 + \frac{\lambda_{\min}}{\lambda_{\max}})^T \|\widetilde{\theta} - \theta\|_2).$$

$\square$

**Proof for Thm. 4.2**  As we discuss in Example. D.3, Assump. C is satisfied with $\mu = 1$. The proof is a direct application of Thm. D.8 with the same choice of dual variables as Eq. (7).

# E  MISSING PROOFS FOR EXISTENCE WHEN THE TRUE MODEL $f^*$ IS UNKNOWN

In the following, we establish some technical lemmas for the maximal likelihood estimator. Given a steering dynamics model class $\mathcal{F}$ and the true dynamics $f^* \sim p_0$ and a steering strategy $\psi : \Pi \to \mathcal{U}$, we consider a steering trajectory $\tau_{T_0} := \{\boldsymbol{\pi}_1, \boldsymbol{u}_1, ..., \boldsymbol{\pi}_{T_0}, \boldsymbol{u}_{T_0}, \boldsymbol{\pi}_{T_0+1}\}$ generated by:

$$\forall t \in [T_0], \quad \boldsymbol{u}_t \leftarrow \psi(\boldsymbol{\pi}_t), \ \boldsymbol{\pi}_{t+1} \sim f^*(\cdot|\boldsymbol{\pi}_t, \boldsymbol{u}_t), \tag{9}$$

where $\boldsymbol{\pi}_{t+1}$ is independent w.r.t. $\boldsymbol{\pi}_{t'}$ for $t' < t$ conditioning on $\boldsymbol{\pi}_t$. In the following, we will denote $\tau_t := \{\boldsymbol{\pi}_1, \boldsymbol{u}_1, ..., \boldsymbol{\pi}_t, \boldsymbol{u}_t, \boldsymbol{\pi}_{t+1}\}$ to be the trajectory up to step $t$.

For any $f \in \mathcal{F}$, we define:

$$p_f(\tau_{T_0}) := \prod_{t=1}^{T_0} f(\boldsymbol{\pi}_{t+1}|\boldsymbol{\pi}_t, \boldsymbol{u}_t). \tag{10}$$

Given $\tau_{T_0}$, we use $\bar{\tau}_{T_0}$ to denote the "tangent" trajectory $\{(\boldsymbol{\pi}_t, \boldsymbol{u}_t, \bar{\boldsymbol{\pi}}_{t+1})\}_{t=1}^{T_0}$ where $\bar{\boldsymbol{\pi}}_{t+1} \sim f^*(\cdot|\boldsymbol{\pi}_t, \boldsymbol{u}_t)$ is independently sampled from the same distribution as $\boldsymbol{\pi}_{t+1}$ conditioning on the same $\boldsymbol{\pi}_t$ and $u_t$.

**Lemma E.1.** *Let $l : \Pi \times \mathcal{U} \times \Pi \to \mathbb{R}$ be a real-valued loss function. Define $L(\tau_{T_0}) := \sum_{t=1}^{T_0} l(\boldsymbol{\pi}_t, \boldsymbol{u}_t, \boldsymbol{\pi}_{t+1})$ and $L(\bar{\tau}_{T_0}) := \sum_{t=1}^{T_0} l(\boldsymbol{\pi}_t, \boldsymbol{u}_t, \bar{\boldsymbol{\pi}}_{t+1})$. Then, for arbitrary $t \in [T_0]$,*

$$\mathbb{E}[\exp(L(\tau_t) - \log \mathbb{E}_{\bar{\tau}_{T_0}}[\exp(L(\bar{\tau}_t))|\tau_t])] = 1.$$

*Proof.* We denote $E^i := \mathbb{E}_{\bar{\pi}_{i+1}}[\exp(l(\pi_i, u_i, \bar{\pi}_{i+1}))|\pi_i, u_i, f^*]$. By definition, we have:

$$\mathbb{E}_{\bar{\tau}_t}[\exp(\sum_{i=1}^{t} l(\pi_i, u_i, \bar{\pi}_{i+1}))|\tau_t] = \prod_{i=1}^{k} E^i.$$

Therefore,

$$\mathbb{E}_{\tau_{T_0}}[\exp(L(\tau_{T_0}) - \log \mathbb{E}_{\bar{\tau}_{T_0}}[\exp(L(\bar{\tau}_{T_0}))|\tau_{T_0}])]$$

$$=\mathbb{E}_{\tau_{T_0-1} \cup \{\pi_{T_0}, u_{T_0}\}}[\mathbb{E}_{\pi_{T_0+1}}[\frac{\exp(\sum_{t=1}^{T_0} l(\pi_t, u_t, \pi_{t+1}))}{\mathbb{E}_{\bar{\tau}_{T_0}}[\exp(\sum_{t=1}^{T_0} l(\pi_t, u_t, \pi_{t+1}))|\tau_{T_0}]}|\tau_{T_0-1} \cup \{\pi_{T_0}, u_{T_0}\}]]$$

$$=\mathbb{E}_{\tau_{T_0-1} \cup \{\pi_{T_0}, u_{T_0}\}}[\mathbb{E}_{\pi_{T_0+1}}[\frac{\exp(\sum_{t=1}^{T_0} l(\pi_t, u_t, \pi_{t+1}))}{\prod_{t=1}^{T_0} E^t}|\tau_{T_0-1} \cup \{\pi_{T_0}, u_{T_0}\}]]$$

$$=\mathbb{E}_{\tau_{T_0-1} \cup \{\pi_{T_0}, u_{T_0}\}}[\frac{\exp(\sum_{t=1}^{T_0-1} l(\pi_t, u_t, \pi_{t+1}))}{\prod_{t=1}^{T_0-1} E^t} \cdot \mathbb{E}_{\pi_{T_0+1}}[\frac{l(\pi_{T_0}, u_{T_0}, \pi_{T_0+1})}{E^{T_0}}|\tau_{T_0-1} \cup \{\pi_{T_0}, u_{T_0}\}]]$$

$$=\mathbb{E}_{\tau_{T_0-1}}[\frac{\exp(\sum_{t=1}^{T_0-1} l(\pi_t, u_t, \pi_{t+1}))}{\prod_{t=1}^{T_0-1} E^t}] = ... = 1.$$

$\square$

**Lemma E.2.** *[Property of the MLE Estimator] Under the condition in Prop. 4.4, given the true model $f^*$ and any deterministic steering strategy $\psi : \Pi \to \mathcal{U}$, define $f_{MLE} \leftarrow \arg\max_{f \in \mathcal{F}} \sum_{t=1}^{T_0} \log f(\pi_{t+1}|\pi_t, u_t)$, where the trajectory is generated by:*

$$\forall t \in [T_0], \quad u_t \leftarrow \psi(\pi_t), \pi_{t+1} \sim f^*(\cdot|\pi_t, u_t),$$

*then, for any $\delta \in (0, 1)$, w.p. at least $1 - \delta$, we have:*

$$\sum_{t=1}^{T_0} \mathbb{H}^2(f_{MLE}(\cdot|\pi_t, u_t), f^*(\cdot|\pi_t, u_t)) \leqslant \log(\frac{|\mathcal{F}|}{\delta}).$$

*Proof.* Given a model $f \in \mathcal{F}$, we consider the loss function:

$$l_M(\pi, u, \pi') := \begin{cases} \frac{1}{2} \log \frac{f(\pi'|\pi, u)}{f^*(\pi'|\pi, u)}, & \text{if } f^*(\pi'|\pi, u) \neq 0 \\ 0, & \text{otherwise} \end{cases}$$

Considering the event $\mathcal{E}$:

$$\mathcal{E} := \{-\log \mathbb{E}_{\bar{\tau}_{T_0}}[\exp L_M(\bar{\tau}_{T_0})|\tau_{T_0}] \leqslant -L_M(\tau_{T_0}) + \log(\frac{|\mathcal{F}|}{\delta}), \quad \forall f \in \mathcal{F}\}.$$

where we define $L_M(\tau_{T_0}) := \sum_{t=1}^{T_0} l_M(\pi_t, u_t, \pi_{t+1})$ and $L_M(\bar{\tau}_{T_0}) := \sum_{t=1}^{T_0} l_M(\pi_t, u_t, \bar{\pi}_{t+1})$. Besides, by applying Lem. E.1 on $l_M$ defined above and applying Markov inequality and the union bound over all $f \in \mathcal{F}$, we have $\Pr(\mathcal{E}) \geqslant 1 - \delta$. On the event $\mathcal{E}$, we have:

$$-\log \mathbb{E}_{\bar{\tau}_{T_0}}[\exp L_{f_{MLE}}(\bar{\tau}_{T_0})|\tau_{T_0}]$$

$$\leqslant -L_{f_{MLE}}(\tau_{T_0}) + \log(\frac{|\mathcal{F}|}{\delta})$$

$$\leqslant l_{MLE}(f^*) - l_{MLE}(f_{MLE}) + \log(\frac{|\mathcal{F}|}{\delta})$$

$$\leqslant \log(\frac{|\mathcal{F}|}{\delta}). \qquad (f_{MLE} \text{ maximizes the log-likelihood})$$

Therefore,

$$\log(\frac{|\mathcal{F}|}{\delta}) \geqslant -\sum_{t=1}^{T_0} \log \mathbb{E}_{\bar{\tau}_{T_0}}[\sqrt{\frac{f(\bar{\pi}_{t+1}|\pi_t, u_t)}{f^*(\bar{\pi}_{t+1}|\pi_t, u_t)}}|\pi_t, u_t, f^*]$$

$$\geqslant \sum_{t=1}^{T_0} 1 - \mathbb{E}_{\bar{\pi}_{t+1}}\left[\sqrt{\frac{f(\bar{\pi}_{t+1}|\pi_t, u_t)}{f^*(\bar{\pi}_{t+1}|\pi_t, u_t)}}|\pi_t, u_t, f^*\right] \qquad (-\log x \geqslant 1 - x)$$

$$= \sum_{t=1}^{T_0} \mathbb{H}^2(f(\cdot|\pi_t, u_t), f^*(\cdot|\pi_t, u_t)).$$

$\square$

**Example 4.4.** [One-Step Difference] If $\forall \pi \in \Pi$, there exists a steering reward $u_\pi \in \mathcal{U}$, s.t. $\min_{f,f' \in \mathcal{F}} \mathbb{H}^2(f(\cdot|\pi, r + u_\pi), f'(\cdot|\pi, r + u_\pi)) \geqslant \zeta$, for some universal $\zeta > 0$, where $\mathbb{H}$ is the Hellinger distance, then for any $\delta \in (0, 1)$, $\mathcal{F}$ is $(\delta, T_\mathcal{F}^\delta)$-identifiable with $T_\mathcal{F}^\delta = O(\zeta^{-1} \log(|\mathcal{F}|/\delta))$.

*Proof.* Consider the steering strategy $\psi(\pi) = u_\pi$. Given any $f \in \mathcal{F}$, and the trajectory sampled by $\psi$ and $f$, by Lem. E.2, w.p. $1 - \frac{\delta}{|\mathcal{F}|}$, if $f_{\text{MLE}} \neq f$, we have:

$$2\log(\frac{|\mathcal{F}|}{\delta}) \geqslant \sum_{t=1}^{T_0} \mathbb{H}^2(f(\cdot|\pi_t, u_t), f_{\text{MLE}}(\cdot|\pi_t, u_t)) \geqslant T_0 \zeta.$$

By union bound over all $f \in \mathcal{F}$, if $T_0 = \lceil \frac{4}{\zeta} \log \frac{|\mathcal{F}|}{\delta} \rceil + 1$, the following holds:

$$\max_{f \in \mathcal{F}} \mathbb{E}_{f, \psi}\left[\mathbb{I}[f = f_{\text{MLE}}]\right] = \max_{f \in \mathcal{F}} \mathbb{E}_{f, \psi}\left[\mathbb{I}[f = \arg\max_{f' \in \mathcal{F}} \sum_{t=1}^{T_\delta} \log f'(\pi_{t+1}|\pi_t, u_t)]\right] \geqslant 1 - \delta.$$

$\square$

**Theorem 4.5.** [A Sufficient Condition for Existence] Given any $\varepsilon > 0$, $\Psi_T^\varepsilon(\mathcal{F}; \pi_1)^5 \neq \varnothing$, if $\exists \tilde{T} < T$, s.t., (1) $\mathcal{F}$ is $(\frac{\varepsilon}{2\eta_{\max}}, \tilde{T})$-identifiable, (2) $\Psi_{T-\tilde{T}}^{\varepsilon/2}(\mathcal{F}; \pi_{\tilde{T}}) \neq \varnothing$ for any possible $\pi_{\tilde{T}}$ generated at step $\tilde{T}$ during the steering.

*Proof.* We denote $\psi_{\text{Explore}} := \{\psi_{\text{Explore}, t}\}_{t \in [T]}$ to be the exploration strategy to identify $f^*$. Given a $\pi_{\tilde{T}}$, we denote $\psi_{\pi_{\tilde{T}}}^{\varepsilon/2} := \{\psi_{\pi_{\tilde{T}}, t}^{\varepsilon/2}\}_{t \in [T]} \in \Psi_{T-\tilde{T}}^{\varepsilon/2}(\pi_{\tilde{T}})$ to be one of the steering strategy with $\varepsilon$-optimal gap starting from $\pi_{\tilde{T}}$.

We consider the history-dependent steering strategy $\psi := \{\psi_t\}_{t \in [T]}$, such that for $t \leqslant \tilde{T}$, $\psi_t = \psi_{\text{Explore}, t}$, and for all $t > \tilde{T}$, we have $\psi_t = \psi_{\pi_{\tilde{T}}, t}^{\varepsilon/2}$.

As a result, for any $f \in \mathcal{F}$, the final gap would be:

$$\Delta_{\psi, T}(f) = \Pr(f_{\text{MLE}} = f) \cdot \frac{\varepsilon}{2} + \Pr(f_{\text{MLE}} \neq f) \cdot \eta_{\max} \leqslant \varepsilon,$$

which implies $\psi \in \Psi_T^\varepsilon(\mathcal{F}; \pi_1)$.

$\square$

# F   GENERALIZATION TO PARTIAL OBSERVATION MDP SETUP

## F.1   POMDP BASICS

**Partial Observation Markov Decision Process**   A (finite-horizon) Partial-Observation Markov Decision Process (with hidden states) can be specified by a tuple $M := \{\nu_1, T, \mathcal{X}, \mathcal{U}, \mathcal{O}, \mathbb{T}, \eta, \mathbb{O}\}$. Here $\nu_1$ is the initial state distribution, $L$ is the maximal horizon length, $\mathcal{X}$ is the hidden state space, $\mathcal{U}$ is the action space, $\mathcal{O}$ is the observation space. Besides, $\mathbb{T} : \mathcal{X} \times \mathcal{U} \to \mathcal{X}$ denotes the stationary transition function, $\mathbb{O} : \mathcal{X} \to \Delta(\mathcal{O})$ denotes the stationary emission model, i.e. the probability of some observation conditioning on some state. We will denote $\mathcal{H}_h := \mathcal{O}_1 \times \mathcal{U}_1 ... \times \mathcal{O}_h$ to be the history space, and use $\tau_h := \{o_1, u_1, ..., o_h\}$ to history observation up to step $h$. We consider the history dependent policy $\psi := \{\psi_1, ..., \psi_H\}$ with $\psi_h : \mathcal{H}_h \to \Delta(\mathcal{U})$. Starting from the initial state $x_1$, the trajectory induced by a policy $\psi$ is generated by:

$$\forall h \in [H], \quad o_h \sim \mathbb{O}(\cdot|x_h), \quad u_h \sim \psi_h(\cdot|\tau_h), \quad \eta_h \sim \eta_h(o_h, u_h), \quad x_{h+1} \sim \mathbb{T}(\cdot|x_h, u_h).$$

---

[5]Here we highlight the dependence on initial policy, model, and time for clarity (see Footnote 2)

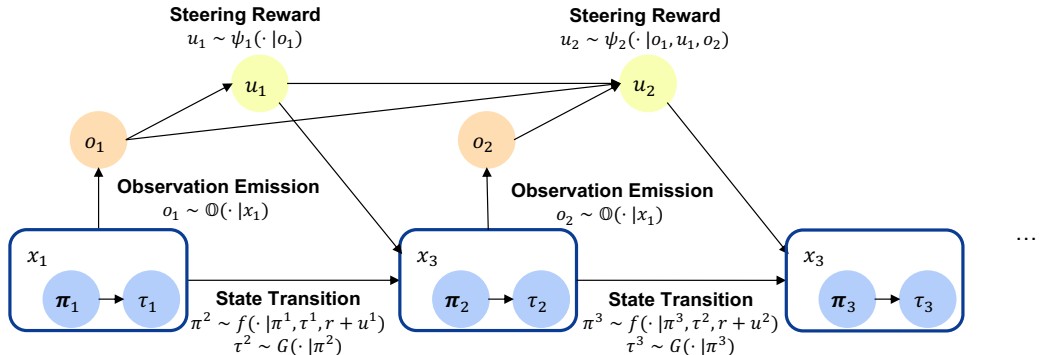

Figure 4: **Probabilistic Graphic Model (PGM) of the POMDP formulation of the steering process**. Starting with the initial state $x_1 := (\pi_1, \tau_1)$, for all $t \geqslant 1$, the mediator receives observation $o_t \sim \mathbb{O}(\cdot|x_t)$ and output the steering reward given the history $u_t \sim \psi(\cdot|o_1, u_1, ..., o_t)$. The agents then update their policies following the dynamics $f$ and the modified reward function $r + u_t$.

### F.2 Steering Process as a POMDP

Given a game $G$, we consider the following Markovian agent dynamics:

$$\forall t \in [T], \quad \tau_t \sim \boldsymbol{\pi}_t, \quad \boldsymbol{\pi}_{t+1} \sim f(\cdot|\boldsymbol{\pi}_t, \tau_t, r),$$

where $\tau_t := \{s_1^{t,k}, a_1^{t,k}, ,..., s_H^{t,k}, a_H^{t,k}\}_{k=1}^K$ is several trajectories generated by the policy $\boldsymbol{\pi}_t$.

In each step $t$, we assume the agents first collect trajectories $\tau_t$ with policy $\boldsymbol{\pi}_t$, and then optimize their policies following some update rule $f(\cdot|\boldsymbol{\pi}_t, \tau_t, r)$. Comparing with the Markovian setup in Sec. 3, here $f$ has additional dependence on the trajectories $\tau_t$.

Based on this new formulation, the dynamics given the steering strategy is defined by:

$$\forall t \in [T], \quad \tau_t \sim \boldsymbol{\pi}_t, \quad u_t \sim \psi_t(\cdot|\tau_1, u_1, ..., \tau_{t-1}, u_{t-1}, \boldsymbol{\pi}_t), \quad \boldsymbol{\pi}_{t+1} \sim f(\cdot|\boldsymbol{\pi}_t, r + u_t),$$

In Fig. 4, we illustrate the steering dynamics by Probabilistic Graphical Model (PGM). Here we treat the joint of $\boldsymbol{\pi}_t$ and $\tau_t$ as the hidden state at step $t$, and the trajectory $\tau_t$ is the partial observation $o_t$ received by the mediator. Next, we introduce the notion of decodable POMDP, where the hidden state is determined by a short history.

**Definition F.1** ($m$-Decodable POMDP). Given a POMDP $M$, we say it is $m$-decodale, if there exists a decoder $\phi$, such that, $x_h = \phi(o_{h-m}, u_{h-m}, ...o_{h-1}, u_{h-1}, o_h)$,

In our steering setting, if for any $f \in \mathcal{F}$, $f$ is $m$-decodable, we just need to learn a steering strategy $\psi := (\mathcal{O} \times \mathcal{U})^m \times \mathcal{O} \to \mathcal{U}$, which predicts the steering reward given the past $m$-step history. This is the motivation for our experiment setup in the Grid World Stag Hunt game in Sec. 6.1. More concretely, we assume the agents trajectories in the past few steps can be used as sufficient statistics for the current policy, and use them as input of the steering strategy (see Appx. G.2.2 for more details).

## G Missing Experiment Details

### G.1 About Initialization in Evaluation

In some experiments, we will evaluate our steering strategies with multiple different initial policy $\boldsymbol{\pi}_1$, in order to make sure our evaluation results are representative.

Here we explain how we choose the initial policies $\boldsymbol{\pi}_1$. We will focus on games with two actions which is the only case we use this kind of initialization. For each player, given an integer $i$, we construct an increasing sequence with common difference $\text{Seq}_i := (\frac{1}{2i}, \frac{3}{2i}, ..., \frac{2i-1}{2i})$. Then, we consider the initial policies $\boldsymbol{\pi}_1$ such that $\pi^1(a^1) = 1 - \pi^1(a^2) \in \text{Seq}_i$, $\pi^2(a^1) = 1 - \pi^2(a^2) \in \text{Seq}_i$. In this way, we obtain a set of initial policies uniformly distributed in grids with common difference

$\frac{1}{i}$. As a concrete example, the initial points in Fig. 1-(b) marked in color black is generated by the above procedure with $i = 10$.

## G.2 EXPERIMENTS FOR KNOWN MODEL SETTING

### G.2.1 EXPERIMENT DETAILS IN NORMAL-FORM STAG HUNT GAME

We provide the missing experiment details for the steering experiments in Fig. 1-(b).

**Choice of $\eta^{\text{goal}}$**  We consider the total utility as the goal function. But for the numerical stability, we choose $\eta^{\text{goal}}(\boldsymbol{\pi}) = \sum_{n \in [N]} J^n_{|\boldsymbol{r}}(\boldsymbol{\pi}) - 10$ where we shift the reward via the maximal utility value 10.

**The Steering Strategy**  The steering strategy is a 2-layer MLP with 256 hidden layers and `tanh` as the activation function. Given a time step $t$ and the policy $\boldsymbol{\pi}_t := \{\pi^1_t, \pi^2_t\}$ with $\pi^n_t(\text{H}) + \pi^n_t(\text{G}) = 1$ for $n \in \{1, 2\}$, the input of the steering strategy is

$$(\log \sqrt{\frac{\pi^1_t(\text{H})}{\pi^1_t(\text{G})}}, -\log \sqrt{\frac{\pi^1_t(\text{H})}{\pi^1_t(\text{G})}}, \log \sqrt{\frac{\pi^2_t(\text{H})}{\pi^2_t(\text{G})}}, -\log \sqrt{\frac{\pi^2_t(\text{H})}{\pi^2_t(\text{G})}}, \frac{T-t}{100}). \tag{11}$$

Here the first (second) two components correspond to the "dual variable" of the policy $\pi^1_t(\text{H})$ and $\pi^1_t(\text{G})$ ($\pi^2_t(\text{H})$ and $\pi^2_t(\text{G})$), respectively; the last component is the time embedding because our steering strategy is time-dependent.

The steering strategy will output a vector with dimension 4, which corresponds to the steering rewards for two actions of two players. Note that here the steering reward function $u^1 : \mathcal{S} \times \mathcal{A}^1[0, U_{\max}]$ (for agent 1) and $u^2 := \mathcal{S} \times \mathcal{A}^2 \to [0, U_{\max}]$ (agent 2) is defined on the joint of state space and individual action space. This can be regarded as a specialization of the setup in our main text, where we consider $u^n : \mathcal{S} \times \mathcal{A} \to [0, U_{\max}] \; \forall n \in [N]$, which is defined on the joint of state space and the entire action space.

**Training Details**  The maximal steering reward $U_{\max}$ is set to be 10, and we choose $\beta = 25$. We use the PPO implementation of StableBaseline3 (Raffin et al., 2021). The training hyper-parameters can be found in our codes in our supplemental materials.

During the training, the initial policy is randomly selected from the feasible policy set, in order to ensure the good performance in generalizing to unseen initialization points. Another empirical trick we adopt in our experiments is that, we strengthen the learning signal of the goal function by including $\eta^{\text{goal}}(\boldsymbol{\pi}_t)$ for each step $t \in [T]$. In another word, we actually optimize the following objective function:

$$\psi^* \leftarrow \arg \max_{\psi \in \Psi} \frac{1}{|\mathcal{F}|} \sum_{f \in \mathcal{F}} \mathbb{E}_{\psi, f} \Big[ \beta \cdot \eta^{\text{goal}}(\boldsymbol{\pi}_{T+1}) + \sum_{t=1}^{T} \beta \cdot \eta^{\text{goal}}(\boldsymbol{\pi}_t) - \eta^{\text{cost}}(\boldsymbol{\pi}_t, \boldsymbol{u}_t) \Big]. \tag{12}$$

The main reason is that here $T = 500$ is very large, and if we only have the goal reward at the terminal step, the learning signal is extremely sparse and the learning could fail.

**Other Experiment Results**  In Fig. 5, we investigate the trade-off between steering gap and the steering cost when choosing different coefficients $\beta$. In general, the larger $\beta$ can result in lower steering gap and higher steering cost.

### G.2.2 EXPERIMENT DETAILS IN GRID-WORLD VERSION OF STAG HUNT GAME

We recall the illustration in LHS of Fig. 2. We consider a 3x3 grid world environment with two agents (blue and red). At the bottom-left and up-right blocks, we have 'stag' and 'hares', respectively, whose positions are fixed during the game. At the beginning of each episode, agents start from the up-left and bottom-right blocks, respectively.

For each time step $h \in [H]$, every agent can take four actions {up, down, left, right} to move to the blocks next to their current blocks. But if the agent hits the wall after taking the action (e.g. the agent locates at the most right column and takes the action right), it will not move. As long as one

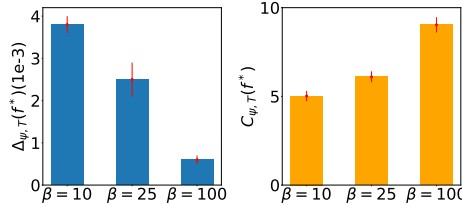

Figure 5: Trade-off between Steering Gap (Left) and Steering Cost (Right). (averaged over 5x5 uniformly distributed grids as initializations of $\pi_1$, see Appx. G.1).

agent reaches the block with either stag or hare, the agents will receive rewards and be reset to the initial position (up-left and bottom-right blocks). The reward is defined by the following.

- If both agents reach the block with stag at the same time, each of them receive reward 0.25.
- If both agents reach the block with hares at the same time, each of them receive reward 0.1.
- If one agent reaches the block with hares, it will get reward 0.2 and the other get reward 0.
- In other cases, the agents receive reward 0.

We choose $H = 16$. The best strategy is that all the agents move together towards the block with Stag, so within one episode, the agents can reach the Stag 16 / 2 = 8 times, and the maximal total return would be 8 * 0.25 = 4.0.

In the following, we introduce the training details. Our grid-world environment and the PPO training algorithm is built based on the open source code from (Lu et al., 2022).

**Agents Learning Dynamics**   The agents will receive a 3x3x4 image encoding the position of all objects to make the decision. The agents adopt a CNN, and utilize PPO to optimize the CNN parameters with learning rate 0.005.

**Steering Setup and Details in Training Steering Strategy**   Our steering strategy is another CNN, which takes the agents recent trajectories as input. More concretely, for each steering iteration $t$, we ask the agents to interact and generated 256 episodes with length $H$, and concatenate them together to a tensor with shape [256 * $H$, 3, 3, 4]. The mediator takes that tensor as input and output an 8-dimension steering reward vector. Here the steering rewards corresponds to the additional rewards given to the agents when one of them reach the blocks with stag or hares (we do not provide individual incentives for states and actions before reaching those blocks). To be more concrete, the 8 rewards correspond to the additional reward for blue and red agents for the following 4 scenarios: (1) both agents reach stag together (2) both agents reach hares together (3) this agent reach stag while the other does not reach stag (4) this agent reach hares while the other does not reach hares.

The steering strategy is also trained by PPO. We choose $\beta = 25$ and learning rate 0.001. We consider the total utility as the goal function, and we adopt the similar empirical trick as the normal-form version, where we include $\eta^{\text{goal}}$ into the reward function for every $t \in [T]$ (Eq. (12)). The results in Fig. 2 is the average of 5 steering strategies trained by different seeds for 80 iterations. The two-sigma error bar is shown.

### G.2.3   EXPERIMENTS IN MATCHING PENNIES

Matching Pennies is a two-player zero-sum game with two actions H=Head and T=Tail and its payoff matrix is presented in Table 2.

**Choice of $\eta^{\text{goal}}$**   In this game, the unique Nash Equilibrium is the uniform policy $\pi^{\text{NE}}$ with $\pi^{n,\text{NE}}(\text{H}) = \pi^{n,\text{NE}}(\text{T}) = \frac{1}{2}$ for all $n \in \{1, 2\}$. We consider the distance with $\pi^{\text{NE}}$ as the goal function, i.e. $\eta^{\text{goal}} = -\|\pi - \pi^{\text{NE}}\|_2$.

**Experiment Setups**   We follow the same steering strategy and training setups for Stag Hunt Game in Appx. G.2.1. The agents follow NPG to update the policies with learning rate $\alpha = 10$.

Table 2: Payoff Matrix of Two-Player Game Matching Pennies. Two actions `H` and `T` stand for `Head` and `Tail`, respectively.

|   | H | T |
|---|---|---|
| H | (1, -1) | (-1, 1) |
| T | (-1, 1) | (1, -1) |

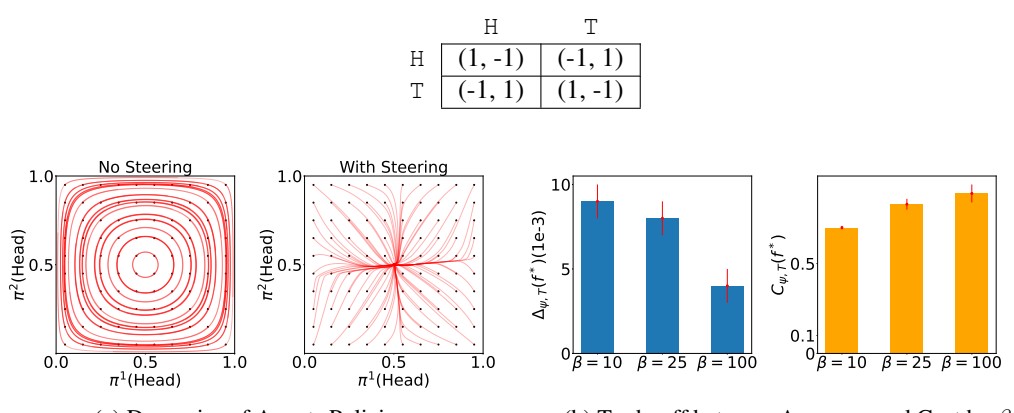

(a) Dynamics of Agents Policies.  (b) Trade-off between Accuracy and Cost by $\beta$.

Figure 6: Experiments in MatchingPennies. (a) $x$ and $y$ axes correspond to the probability that agents take `Head`. Black dots mark the initial policies, and red curves represents the trajectories of agents policies. The steering strategy to plot the figure is trained with $\beta = 25$. (b) We compare $\beta = 10, 25, 100$. Error bar shows 95% confidence intervals. (averaged over 5x5 uniformly distributed grids as initializations of $\boldsymbol{\pi}_1$, see Appx. G.1)

**Experiment Results**  As shown in Fig. 6-(a), we can observe the cycling behavior without steering guidance (Akin and Losert, 1984; Mertikopoulos et al., 2018). In contrast, our learned steering strategy can successfully guide the agents towards the desired Nash. In Fig. 6-(b), we also report the trade-off between steering gap and steering cost with different choice of $\beta$.

### G.3 EXPERIMENTS FOR UNKNOWN MODEL SETTING

#### G.3.1 DETAILS FOR EXPERIMENTS WITH SMALL MODEL SET $\mathcal{F}$

The results in Table 1 is averaged over 5 seeds and the error bars show 95% confidence intervals.

**Training Details for $\psi^*_{0.7}$ and $\psi^*_{1.0}$**  The training of $\psi^*_{0.7}$ and $\psi^*_{1.0}$ follow the similar experiment setup as Appx. G.2.2, except here the agents adopt random learning rates. For the choice of $\beta$, we train the optimal steering strategy with $\beta \in \{10, 20, 30, 40, 50, 60, 70, 80, 90, 100\}$ for both $f_{0.7}$ and $f_{1.0}$, and choose the minimal $\beta$ such that the resulting steering strategy can achieve almost 100% accuracy (i.e. $\Delta_{\psi, f} \leqslant \varepsilon$ for almost all 5x5 uniformly distributed initial policies generated by process in Appx. G.1). As we reported in the main text, we obtain $\beta = 70$ for $f_{0.7}$ and $\beta = 20$ for $f_{1.0}$.

**Training Details for $\psi^*_{\mathbf{Belief}}$**  For the training of $\psi^*_{\text{Belief}}$, the input of the steering strategy is the original state (Eq. (11)) appended by the belief state of the model. In each steering step $t \in [T]$, we assume the mediator can observe a learning rate sample $\alpha$, and use it to update the model belief state correspondingly. The regularization coefficient $\beta$ for the training of $\psi_{\text{Belief}}$ is set to be the expected regularization coefficient over the belief state $\beta = b(f_{0.7}) \cdot 70 + b(f_{1.0}) \cdot 20$. In another word, we use the sum of the coefficient of two models weighted by the belief state. This is reasonable by the definition of the reward function in the belief state MDP lifted from the original POMDP. $\psi^*_{\text{Belief}}$ is trained the PPO algorithm.

During the training of $\psi^*_{\text{Belief}}$, we find that the train is not very stable, possibly because the chosen $\beta$ for two models are quite different. Therefore, we keep tracking the steering gap of the steering strategy during the training and save the model as long as it outperforms the previous ones in steering gap. Our final evaluation is based on that model.

### G.3.2 DETAILS FOR EXPERIMENTS WITH LARGE MODEL SET $\mathcal{F}$

We set $U_{\max} = 1.0$, and the random exploration strategy (red curve in the left sub-plot in Fig. 3) will sample the steering reward uniformly from the interval $[0, U_{\max}]$. We use the PPO (Raffin et al., 2021) to train of exploration policy and also the steering strategy given hidden model. To amplify the exploration challenge, we set $\beta^n = 1$ when $\pi^n(\mathbb{A}) \leqslant 0.5$ and increase to $\beta^n = 10$ when $\pi^n(\mathbb{A}) \leqslant 0.5$. As a result, if the mediator follows first-explore-then-exploit strategy and fail to distinguish avaricious agents from the normal ones, adopting large steering reward can lead to much worse performance.

For the training of exploration policy, although the learning signal $\mathbb{I}[f = f_{\mathrm{MLE}}]$ in Proc. 2 is supported by theory, it contains much less information than the posterior probability $[\Pr(f|\boldsymbol{\pi}_1, u_1, ..., \boldsymbol{\pi}_T, u_T, \boldsymbol{\pi}_{T+1})]_{f \in \mathcal{F}}$. Therefore, empirically, we instead train a history-independent steering strategy to maximize the posterior probability of $f$:

$$\psi^{\text{Explore}} \leftarrow \arg\max_{\psi} \frac{1}{|\mathcal{F}|} \sum_{f \in \mathcal{F}} \mathbb{E}_{\psi, f} \Big[ \sum_{n \in [N]} \Pr(\lambda^n | \boldsymbol{\pi}_1, u_1, ..., \boldsymbol{\pi}_{\widetilde{T}}, u_{\widetilde{T}} \boldsymbol{\pi}_{\widetilde{T}+1}) \Big]. \tag{13}$$

Here we use the sum of posteriors of $\lambda^n$s since the $\lambda^n$s are independent for all $n \in [N]$. We observe it results in better performance, and it is doable by keep tracking the model belief state of each agent. Besides, similar to Stag Hunt games, we observe that using the posterior $\Pr(\lambda^n | \boldsymbol{\pi}_1, u_1, ..., \boldsymbol{\pi}_t, u_t, \boldsymbol{\pi}_{t+1})$ as rewards in the non-terminal steps $t < T$ increase the performance, and we use the same trick (Eq. (12)).

To plot the results in the middle and right sub-plots in Fig. 3, for each model $f^* \in \{f_1, f_2, f_3\}$, we train three steering strategies (with the same state design in Eq. (11)). The first one is the oracle strategy, which starts with $\boldsymbol{\pi}_1$ and steering for $T = 500$ steps. The second one is the FETE strategy, including an exploration policy and another exploitation policy. The exploration policy is trained following the above Eq. (13) with exploration horizon $\widetilde{T} = 30$. Then, we estimate the model from samples generated by the exploration policy, and train another exploitation policy following the remaining step of FETE with exploitation horizon $T = 470$ (Procedure 2). The third strategy FETE-RE is the same as FETE except we just use random policy as the exploration policy, and estimate the model by interaction samples generated by that. During the evaluation, for FETE and FETE-RE, we steer with the exploration policy for the first 30 steps, and execute the exploitation policy for the rest. The results are averaged over 5 seeds and two-sigma error bar is shown.

### G.4 EXPLANATION OF THE CONSISTENCY OF THE ADAPTION

We first want to highlight it is not easy to have an apples-to-apples comparison with (Canyakmaz et al., 2024). First, because the experiment setting in (Canyakmaz et al., 2024) does not present significant exploration challenges, we design and decide to evaluate both methods in the "avaricious agents" setting (Fig. 3). Second, SIAR-MPC is specialized for polynomial function classes and the dynamics tractable by MPC, making it difficult to generalize beyond that setting. Therefore, we have to do some necessary adaption. To ensure the fair comparison, we consider to use FETE-RE as the adaption of SIAR-MPC in our setting. For the exploration stage, FETE-RE aligns with SIAR-MPC in using random exploration. For the model identification and exploitation phase, FETE-RE adopts MLE estimation and RL methods to train the exploitation policy, which inherit the same inspirits as SIAR-MPC and also aligned with our original FETE.

From another perspective, the main focus of our empirical comparison between FETE and (Canyakmaz et al., 2024) is the impact of different exploration strategies on the final steering gap and steering costs. This is reasonable. Because from the discussion in Sec. 5.2, we can conclude that our FETE is more general compared with SIAR-MPC in (Canyakmaz et al., 2024) in terms of both the model estimation strategy and exploitation strategy. The main distinction and improvement of our FETE compared with (Canyakmaz et al., 2024) is our strategic exploration strategy.

### G.5 A SUMMARY OF THE COMPUTE RESOURCES BY EXPERIMENTS IN THIS PAPER

**Experiments on Two-Player Normal-Form Games** For the experiments in 'Stag Hunt' and 'Matching Pennies' (illustrated in Fig. 1, 5, 6), we only use CPUs (AMD EPYC 7742 64-Core Processor). It takes less than 5 hours to finish the training.

**Experiments on Grid-World Version of 'Stag Hunt'**    For the experiments in grid-world 'Stag Hunt' (illustrated in Fig. 2), we use one RTX 3090 and less than 5 CPUs (AMD EPYC 7742 64-Core Processor). The training (per seed) takes around 48 hours.

**Experiments on $N$-Player Normal-Form Cooperative Games**    For the experiments in cooperative games (illustrated in Fig. 3), we only use CPUs (AMD EPYC 7742 64-Core Processor). It takes less than 10 hours to finish the training.

# H    ADDITIONAL DISCUSSION ABOUT GENERALIZING OUR RESULTS

In this section, we discuss some extensions of the principle and algorithms in this paper to more general settings.

**Non-Tabular Setting**    When the game is non-tabular and its state and action spaces are infinite, the steering problem itself is fundamentally challenging without additional assumptions, since the policies are continuous distributions with infinite dimension and the learning dynamics can be arbitrarily complex. Nonetheless, when the agents' policies and steering rewards are parameterized by finite variables, our methods and algorithms can still be generalized by treating the parameters as representatives.

As a concrete example, the Linear-Quadratic (LQ) game is a popular model involving countinuous state and action spaces (Jacobson, 1973; Başar and Bernhard, 2008; Zhang et al., 2019). In zero-sum LQ game, the game dynamics are characterized by a linear system:

$$x_{t+1} = Ax_t + By_t + Cz_t,$$

with one-step reward function

$$r^1(x_t, y_t, z_t) = -r^2(x_t, y_t, z_t) = x_t^\top Q x_t + y_t^\top R^u y_t - z_t^\top R^v z_t.$$

Here $x_t, x_{t+1} \in \mathbb{R}^d$ are the system states, $y_t \in \mathbb{R}^{m_1}$ and $z_t \in \mathbb{R}^{m_2}$ denote the actions of two agents.

Besides, the agents policies are parameterized by matrices $K_t \in \mathbb{R}^{m_1 \times d}, L_t \in \mathbb{R}^{m_2 \times d}$, i.e.

$$y_t = -K_t x_t, \quad z_t = -L_t x_t.$$

Following the quadratic form of the original reward, one may consider quadratic steering reward functions with parameters $\Theta_t := (\Theta_t^Q, \Theta_t^u, \Theta_t^v)$ and $\Xi_t := (\Xi_t^Q, \Xi_t^u, \Xi_t^v)$, such that, the steering reward for two agents at step $t$ is specified by:

$$u_t^1(x_t, y_t, z_t) = x_t^\top \Theta_t^Q x_t + y_t^\top \Theta_t^u y_t - z_t^\top \Theta_t^v z_t,$$
$$u_t^2(x_t, y_t, z_t) = x_t^\top \Xi_t^Q x_t + y_t^\top \Xi_t^u y_t - z_t^\top \Xi_t^v z_t,$$

and the reward after modification would be:

$$r^1(x_t, y_t, z_t) + u_t^1(x_t, y_t, z_t) = x_t^\top(\Theta_t^Q + Q)x_t + y_t^\top(\Theta_t^u + R^u)y_t - z_t^\top(\Theta_t^v + R^v)z_t,$$
$$r^2(x_t, y_t, z_t) + u_t^2(x_t, y_t, z_t) = x_t^\top(\Xi_t^Q - Q)x_t + y_t^\top(\Xi_t^u - R^u)y_t - z_t^\top(\Xi_t^v - R^v)z_t.$$

Although the state, action and steering reward spaces are continuous, both the policies and steering reward are determined by their parameters. Therefore, the agents' learning dynamics can be modeled by a function $f^*$ mapping between those parameters instead:

$$(K_{t+1}, L_{t+1}) \sim f^*(\cdot | \underbrace{(K_t, L_t)}_{\text{agents' policies}}, \underbrace{(\Theta_t^Q + Q, \Theta_t^u + R^u, \Theta_t^u + R^v, \Xi_t^Q - Q, \Xi_t^u - R^u, \Xi_t^u - R^v)}_{\text{modified rewards}}).$$

Besides, the learning of steering strategy is equivalent to learning a function mapping $\psi$ from parameters in history $\{(K_\tau, L_\tau, \Theta_\tau, \Xi_\tau)\}_{\tau=1}^{t-1} \cup \{(K_t, L_t)\}$ to the next steering reward parameter $(\Theta_t, \Xi_t)$. Since both the policy parameters and steering reward parameters have finite dimension, this problem is tractable under our frameworks.

**Uncountable function class** $\mathcal{F}$    Our results can be extended to cases where the model class $\mathcal{F}$ is infinite but has a finite covering number. We denote $\mathcal{F}_{\varepsilon_0}$ as the $\varepsilon_0$-cover for $\mathcal{F}$, s.t.

$$\forall f \in \mathcal{F}, \quad \exists f' \in \mathcal{F}_{\varepsilon_0}, \text{ s.t. } \max_{\boldsymbol{\pi} \in \Pi, \boldsymbol{u}} \mathbb{TV}\Big(f(\cdot|\boldsymbol{\pi}, \boldsymbol{u}) - f'(\cdot|\boldsymbol{\pi}, \boldsymbol{u})\Big) \leqslant \varepsilon_0.$$

where $\mathbb{TV}$ denotes the total variation distance. If $\mathcal{F}$ is uncoutable but $\mathcal{F}_{\varepsilon_0}$ is finite, we run our algorithms with $\mathcal{F}_{\varepsilon_0}$ instead of $\mathcal{F}$. Under Assump. A, we denote $f_{\varepsilon_0}^* \in \mathcal{F}_{\varepsilon_0}$ is the function $\varepsilon_0$ close to $f^*$. By simmulation lemma, then we have:

$$\big|\mathbb{E}_{\psi,f^*}\big[\eta^{\text{goal}}(\boldsymbol{\pi}_{T+1})\big] - \mathbb{E}_{\psi,f_{\varepsilon_0}^*}\big[\eta^{\text{goal}}(\boldsymbol{\pi}_{T+1})\big]\big| \leqslant T \cdot \varepsilon_0 \cdot \eta_{\max}$$

$$\big|\mathbb{E}_{\psi,f^*}\big[\sum_{t=1}^{T} \eta^{\text{cost}}(\boldsymbol{\pi}_t, \boldsymbol{u}_t)\big] - \mathbb{E}_{\psi,f_{\varepsilon_0}^*}\big[\sum_{t=1}^{T} \eta^{\text{cost}}(\boldsymbol{\pi}_t, \boldsymbol{u}_t)\big]\big| \leqslant T^2 \cdot \varepsilon_0 \cdot \max_{\boldsymbol{\pi}, \boldsymbol{u}} \eta^{\text{cost}}(\boldsymbol{\pi}, \boldsymbol{u}).$$

As we can see, we can still optimize the objective in Eq. (1) with $\mathcal{F}_{\varepsilon_0}$, and then transfer guarantees on steering gap and cost for $f_{\varepsilon_0}^*$ (e.g. the the worst case guarantees in Prop. 3.3) to $f^*$ with additional $O(T^2 \cdot \varepsilon_0)$ errors, which is ignorable when $\varepsilon_0$ is small enough.

**Non-Markovian Learning Dynamics**    In general, non-Markovian learning dynamics is intractable, as implied by the fundamental difficulty in learning optimal policies in POMDPs. However, when some special structures exhibit, our methods for Markovian agents can be generalized. One example is the non-Markovian agents with finite-memory, i.e.,

$$\boldsymbol{\pi}_{t+1} \sim f^*(\cdot|\boldsymbol{\pi}_{t-m+1}, \boldsymbol{r} + \boldsymbol{u}_{t-m+1}, ..., \boldsymbol{\pi}_t, \boldsymbol{r} + \boldsymbol{u}_t).$$

This can be reformulated by a Markovian dynamics

$$\boldsymbol{x}_{t+1} \sim F^*(\cdot|\boldsymbol{x}_t, \boldsymbol{r} + \boldsymbol{u}_t),$$

with the same steering rewards as actions but a new definition of "state": $\boldsymbol{x}_t := \{\boldsymbol{\pi}_{t-m+1}, \boldsymbol{r} + \boldsymbol{u}_{t-m+1}, ..., \boldsymbol{\pi}_{t-1}, \boldsymbol{r} + \boldsymbol{u}_{t-1}, \boldsymbol{\pi}_t\}$. Comparing with Def. 3.1, the dimension of the state space is expanded by $m$ times, which is still tractable for small $m$.

**Neural Networks as Model Class to Approximate Complex** $f^*$    The main principle for choosing $\mathcal{F}$ is to ensure our "realizability" assumption holds with high probability, i.e. the true model $f^* \in \mathcal{F}$. The concrete choice of $\mathcal{F}$ depends on the prior knowledge we have about the agents' learning dynamics. In general, the less prior knowledge we have, the larger $\mathcal{F}$ should be to ensure realizability, and vice versa.

In practice, one "safe-choice" can be consider a class of parameterized neural networks as $\mathcal{F}$, since it has been proven in deep RL and supervised learning literature that neural networks have powerful approximation ability when $f^*$ is potentially very complex. Because in our formulation, we allow the randomness of next policy $\boldsymbol{\pi}_{t+1}$ (instead of a deterministic output) given $\boldsymbol{\pi}_t$ and $\boldsymbol{r} + \boldsymbol{u}_t$, we may consider a neural network taking the concatenation of $\boldsymbol{\pi}_t$, $\boldsymbol{r} + \boldsymbol{u}_t$ and another random Gaussian vector $\xi$ as inputs. Here the noise vector is introduced to model the stochasticity of $\boldsymbol{\pi}_t$.

The parameters of neural networks are in general continous variables, which implies the model class is uncountable. However, if the parameters has bounded value range, we can show the finite covering number on the parameter space. If we consider Lipschitz continuous activation functions (which is most of the cases), it implies the bounded covering number.

Besides, when considering neural networks, the resulting model class $\mathcal{F}$ can be extremely large and the MLE-based strategic exploration in Procedure 2 will be inefficient. We highlight that we design such exploration step in order to align with the main principle: *the algorithm design should be supported by theoretical guarantees on the performance of the learned steering strategy*. This focus on theoretical rigor is the main factor limiting the scalability of our algorithms in more complex settings. Conversely, if we relax the requirements on theoretical guarantees, it is not very challenging to adapt our algorithms to complex scenarios. For example, we can instead consider more scalable exploration methods, such as Random Network Distillation (RND) (Burda et al., 2018) or Bootstrapped DQN (Osband et al., 2016), although without theoretical guarantees.

