# OpenReview forum: "Learning to Steer Markovian Agents under Model Uncertainty"
_ICLR.cc/2025/Conference — ICLR 2025 Poster_

### Official Review · Reviewer_oazJ · 2024-10-30

**Soundness:** 3
**Presentation:** 3
**Contribution:** 3
**Rating:** 8
**Confidence:** 4

**Summary:**

The paper addresses the challenge of steering multi-agent systems towards desired policies without having prior knowledge of the agents' learning dynamics. The focus is on a new class of learning dynamics called Markovian agents. The authors propose a model-based non-episodic reinforcement learning (RL) framework and formulate an optimization objective incorporating history-dependent steering strategies to handle model uncertainty. The paper presents theoretical results and practical algorithms to achieve this steering, validated with empirical studies.

**Strengths:**

*Conceptual Novelty and Relevance*: The paper introduces an approach to steering agents by focusing on Markovian agents with limited cognitive resources.

*Theoretical Contributions*: The authors provide a comprehensive theoretical analysis, including sufficient conditions for achieving low steering gaps and the existence of history-dependent steering strategies.

*Algorithmic Development*: The proposed algorithms (belief-state-based steering strategy for small model classes and a First-Explore-Then-Exploit (FETE) framework for large classes) effectively balance the trade-off between tractability and optimality.

**Weaknesses:**

This is a solid paper with well-grounded contributions. My primary concern, however, is that the current setting is limited to a tabular MDP with a finite model class. While this simplified setting is acceptable for a theoretical paper, the corresponding experiments appear relatively basic. I am curious about whether the proposed Steering Dynamics framework can be extended effectively to more realistic scenarios (or if the Steering Dynamics framework is helpful in any real-world situation?) involving non-tabular MDPs and an infinite model class.

**Questions:**

1. What will happen if the model set $\mathcal{F}$ is uncountably infinite?

2. What will happen for a non-tabular MDP setup?

---

> ### Author Response · Authors · 2024-11-17
> **Response to Reviewer oazJ**
>
> We thank the reviewer for the valuable feedback and for acknowledging our contributions!
>
> In light of your insightful comments and suggestions, we have revised the paper to include additional discussion in Appx. H. Below we provide a brief summary of our updates.
>
> ###  1. Extension to uncountable model class $\mathcal{F}$
>
> Our results can be extended to cases where the model class $\mathcal{F}$ is infinite but has a finite covering number. On the algorithmic level, this can be achieved by replacing $\mathcal{F}$ with its $\epsilon_0$-covering set, which is countable. Theoretically, under the realizability assumption (Assump. A), the $\epsilon_0$-covering set of $\mathcal{F}$ will contain at least one element $\epsilon_0$-close to $f^*$. Therefore, the worst case guarantees for $\mathcal{F}$ (e.g. Prop. 3.3) can be transferred to $f^*$ with additional $O(\epsilon_0)$ errors.
>
>
> ### 2. Extension to non-tabular scenarios
>
> When the game is non-tabular and its state and action spaces are infinite, the steering problem itself is fundamentally challenging without additional assumptions, since the policies are continuous distributions with infinite dimension and the learning dynamics can be arbitrarily complex.
>
> Nonetheless, when the agents’ policies and steering rewards are parameterized by finite variables, our methods and algorithms can still be generalized by treating the parameters as representatives. For example, in Linear-Quadratic (LQ) games, the agents’ policies and (steering) rewards are parameterized by finite-dimensional matrices. In that case, the learning dynamics and steering strategies can be simplified as function mappings between those parameters, making them tractable under our current framework. Please refer to Appx. H for more details.

---

> ### Author Response · Authors · 2024-11-22
>
> Dear Reviewer oazJ, thank you again for your time and efforts in the reviewing process and we are happy to address your further questions if any!

---

> ### Comment · Reviewer_oazJ · 2024-11-22
>
> Thanks! The extension makes sense to me. You seem to be using a covering number to measure the uncountable model class, which is exactly what I expected. Anyway, nice work!

---

### Official Review · Reviewer_EzpY · 2024-11-03

**Soundness:** 3
**Presentation:** 1
**Contribution:** 2
**Rating:** 5
**Confidence:** 3

**Summary:**

This paper proposed an RL framework that helps multi-agent systems towards specific behaviors or policies using additional incentives or rewards, even without prior knowledge of agent learning models.

**Strengths:**

Strengths:

1. The authors introduce a new objective function that balances the goal of steering agents toward a target policy while minimizing the overall steering cost.
2. The paper provides sufficient conditions under which the proposed steering strategies are effective, ensuring a low gap between target and actual agent behavior and achieving Pareto optimality.
3. The paper includes experimental results in different environments, demonstrating the algorithms' performance under varying model uncertainty conditions.

**Weaknesses:**

Weakness:

1. Poor writing: The writing in this paper needs significant improvement; it appears to have been put together in a hurry, making it very difficult to read. As a new research problem, the steering problem has not been well-motivated or clearly formulated, leaving key concepts ambiguous. For example, the definitions of "desired policies" and "desired behaviors" are unclear, and it’s difficult to understand their precise meaning in this context. If we already know the desired policies, why any learning is necessary？ Furthermore, the concept of the "mediator" is not introduced or explained and suddenly appears in the paper. Similarly, the notation f∗ is introduced abruptly at line 210 without any prior explanation, creating confusion. The new learning objective presented in line 220 is also hard to understand and lacks sufficient clarification regarding its purpose and relevance.
2. The transitions between paragraphs are also weak, resulting in a lack of cohesion throughout the text. Starting from the problem formulation section, the paper becomes increasingly difficult to follow, with unclear structure and insufficient explanations, making it hard for readers to grasp the main ideas and contributions.

**Questions:**

1. The paper mentions comparison experiments with control methods. In which scenarios does the proposed method outperform control methods, and in which scenarios does it show similar or inferior performance?
2. Could you provide a more precise definition of "desired policies" and "desired behaviors”? What exactly does "desired policy" mean in this paper? Is "desired behavior" different from it?
3. If we already know the desired policy, why is there still a need for learning? Are there certain situations in the steering problem where learning is still required, even if the desired policy is known?
4. Why steering is necessary in multi-agent systems? Can existing algorithms find the equilibrium or approximated equilibrium for MAS?

---

> ### Author Response · Authors · 2024-11-17
> **Response to Reviewer EzpY (Part 1 of 2)**
>
> We regret that you did not enjoy reading the paper as we expected. Below we carefully address your comments in detail. We sincerely hope that our clarifications help enhance the clarity of the paper and address your misunderstandings on our paper, and encourage you to reconsider the evaluation of our work.
>
> ## Response to Weaknesses
>
> ### 1. The Definition of Desired Policies
>
> As clearly explained in line 60, “... desired policies, that is, to minimize the steering gap vis-a-vis the target outcome”. Formally, the “desired policy/outcome” refers to the policy that maximizes the mediator’s goal function $\eta^{goal}$ (the target outcome). **We use the term “desired”, consistent with closely related literature [1,2], because $\eta^{goal}$ can vary a lot depending on the context, resulting in different definitions of what constitutes a favorable behavior.**
>
>
>
> For example, the resulting policy can be extremely different if $\eta^{goal}$ represents the total utility of all agents (social welfare) versus the lowest return among all agents (worst-case utility). Our framework is general, allowing $\eta^{goal}$ to represent any potential objectives. Thus, it is not feasible to provide a specific description for the “desired policy” (e.g. “the welfare-maximizing policy”), and the general term “desired” is more appropriate.
>
> ### 2. Why is learning necessary?
>
> We believe the reviewer may misunderstand what is learned in our steering setting. The uncertainty arises from the unknown agents’ learning dynamics (i.e., $f^*$), which is the target of learning. Even if the desired policy is known, we still need to learn $f^*$, in order to determine how to efficiently steer the agents towards it. Moreover, the desired policy is known only if we have full information about the game and the goal function $\eta^{goal}$. Otherwise, it remains unknown.
>
> Besides, notably, it is a quite strong assumption that the mediator can force the agents to directly take desired policies when they are known. In another word, the mediator can not bypass the learning step by compelling the agents to do what it prefers. Instead, we only assume that the mediator can incentivize the agents by additional payment, which is much weaker and more practical.
>
> ### 3. Others
>
> * The Mediator: We introduce the mediator and explain it in the second paragraph of the introduction. It is a "third-party" outside the game, but can influence the agents by additional steering rewards. The mediator-based steering setup is a standard in the literature [1,2].
>
> * Definition of $f^*$: This notation was first introduced and explained in the introduction (line 93). $f^*$ denotes the true learning dynamics of the agents. In our revision, we make it clear in Assump. A.
>
> * Learning Objective: The purpose of Eq.(1) is to propose a learning objective, and by optimizing it, we can get a good steering strategy. There can be many candidates for the learning objective, we pick Eq.(1) because of its theoretical guarantees (see Prop. 3.3 for more details).

---

> ### Author Response · Authors · 2024-11-17
> **Response to Reviewer EzpY (Part 2 of 2)**
>
> ## Response to Question 1
>
> We would like to clarify the reviewer's misunderstanding regarding the focus of our experiments. **Our goal is not to _"compare with control methods"_, but rather to compare exploration efficiency with concurrent work [1]**. As discussed in the last paragraph in Section 5, our FETE does not impose restrictions on whether control or RL methods are used to compute the steering strategy. The main advantage of our FETE over [1] is that we conduct strategic exploration rather than noise-based exploration in [1]. In experiments, we keep everything the same between FETE and FETE-RE (adaptation of [1] to our setting), except which exploration strategy they use. The key message we want to convey through experiments is that strategic exploration (our FETE) can achieve better performance than random exploration [1] when the exploration challenges present.
>
>
>
> ## Response to Question 2 and 3
>
> Please see the explanation in our responses to weaknesses. In our paper, we use “desired policy”, “desired outcome” or “desired behavior” alternatively to refer to the policy maximizing the mediator’s goal function $\eta^{goal}$.
>
>
>
> ## Response to Question 4
>
> ### 1. Why steering is important?
>
> As discussed at the beginning of the introduction (including Figure 1), self-interested agents may not always converge to the best policies for all of them, potentially resulting in a "lose-lose" outcome. In Figure 1, we illustrate it by agents following replicator dynamics (a standard learning dynamics) in the StagHunt game (a standard example). In this setup, there are two equilibrium policies: (G, G) and (H, H). (H, H) is preferable since each agent receives the highest possible rewards 5, whereas at (G, G), each agent only receives reward 2. However, the agents may converge to (G, G) depending on the initialization. In contrast, with appropriate steering, agents following the original dynamics can be guided to (H, H).
>
>
>
> ### 2. Can existing algorithms find the (approximated) equilibrium for MAS?
>
> Firstly, we believe the reviewer may have fundamentally misunderstood the focus of this paper, as the existence of such an algorithm is unrelated to the topics we address. We clarify it below.
>
> * This paper does not aim to propose new algorithms for agents to solve equilibrium. Instead, we study how to modify the agents’ reward functions, so that **the agents following their original learning dynamics will converge to our desired policies**. Here the mediator has no direct control over the agents’ choices of learning dynamics, but only has limited influence by providing additional incentives.
>
>   Even if learning dynamics existed that can lead agents to the desired policy directly, in practice, the agents may not always adopt them. Our steering framework is particularly valuable for self-incentive-driven and short-sighted agents (as motivated by example in Figure 1), which are common in many scenarios.
>
> * We do not restrict the desired policies to be equilibria. We only assume access to a goal function $\eta^{goal}$, and the desired policy is defined to be its maximizer. For instance, in Prisoner’s Dilemma, if $\eta^{goal}$ is social welfare, the desired policy would be “(Cooperate, Cooperate)”, whereas the unique Nash equilibrium is “(Defect, Defect)”.
>
> Secondly, to answer the reviewer’s question: finding (approximate) equilibrium in MAS is a fundamental question and has been investigated for decades, going back as far as [3]. In general, there does not exist a universal and efficient algorithm which can find Nash equilibrium in all games, and finding Nash equilibrium has been proved to be PPAD-hard [3].
>
>
> [1] Canyakmaz et al., Steering game dynamics towards desired outcomes
>
> [2] Zhang et al., Steering No-Regret Learners to a Desired Equilibrium
>
> [3] Daskalakis et al., The Complexity of Computing a Nash Equilibrium

---

> ### Author Response · Authors · 2024-11-22
>
> Dear Reviewer EzpY, thank you again for your time and efforts in the reviewing process! We hope our responses are helpful, and we are happy to answer your further questions if any. If all of your concerns have been addressed, we will appreciate it if you can let us know whether our response changes your evaluation of our work.

---

> ### Author Response · Authors · 2024-11-29
>
> Dear Reviewer EzpY, given that the discussion period is closing to the end, we would really appreciate it if you can let us know if you have further questions. If all of your concerns are addressed, we kindly request you to reconsider the evaluation of our paper.
>
>
> Thank you once again for your efforts!
>
> Best Regards,
>
> Authors

---

### Official Review · Reviewer_M5wC · 2024-11-04

**Soundness:** 3
**Presentation:** 3
**Contribution:** 2
**Rating:** 6
**Confidence:** 3

**Summary:**

This paper presents a framework for steering the behaviors of agents towards desired outcomes within a reinforcement learning setting, where the exact learning dynamics of the agents are not known. Specifically, this work introduces the concept of Markovian agents, whose decisions are based only on their current states and immediate rewards, and explores the challenges posed by model uncertainty in practical applications. To address these challenges, the study develops a non-episodic RL approach and proposes the First-Explore-Then-Exploit (FETE) framework. This strategy involves initially exploring to identify the most accurate model from a predefined class and then exploiting this model to effectively steer agent behaviors. The theoretical contribution of the paper establishes that if the model class of learning dynamics is identifiable within finite steps, then a steering strategy with an ε-steering gap can exist. The empirical results in the paper demonstrate that their algorithm can operate under the assumption that each player has only two actions.

**Strengths:**

- The method introduced in the paper is commended for its intuitive and logical approach. By assuming that the learning dynamics of the agents are Markovian and can be learned in finite steps, the paper effectively simplifies the complex problem of steering under uncertainty.

- The paper provides transparency regarding the experimental assumptions detailed in the appendices. Moreover, the experimental setup involves a wide range of learning rate combinations, specifically up to 3^10.

- The paper is well-written and very easy to follow, with justification and explanation whenever needed in most places.

**Weaknesses:**

While I appreciate the application of reinforcement learning (RL) to address the steering problem, balancing exploration and exploitation amidst the uncertainty of agents' learning dynamics, but there are some concerns regarding the paper:

- A primary challenge highlighted is the agents' reluctance to disclose their learning dynamics, creating fundamental model uncertainty. The theoretical discussion posits that if the model class $F$ of the agents is identifiable, then an epsilon-steering strategy is viable. It would enhance the paper if the theoretical sections could more explicitly address how understanding the model class $F$ effectively reduces this uncertainty, thus better connecting the theoretical insights with the algorithmic implementations.

- The algorithm proposed in the paper relies heavily on accurately specifying agents' learning dynamics, which presents a significant challenge when considering real-world complexities where agents' behaviors are influenced by others, such as herd behavior or nonconformity. These dynamic inter-agent interactions introduce variability that the algorithm may not capture effectively, given its current design focused on binary or categorical decision-making scenarios. This specificity restricts its utility, particularly in more complex decision environments where outcomes are not merely binary and agent behaviors are interdependent and continuously evolving. Consequently, while the algorithm provides a structured approach to steering agent decisions within a controlled setting, its application to broader, more realistic scenarios involving complex agent dynamics and multiple decision variables may require significant adaptations to accommodate these behavioral complexity.

**Questions:**

- While this paper usefully demonstrates the benefit of steering, it does not sufficiently address whether the proposed steering methods are adequately effective when compared to other potential steering strategies. For example, expanding the comparative analysis to include non-Markovian agent models could help justify the emphasis on Markovian agents in the introduction. Non-Markovian models, which account for past states and decisions in their decision-making processes, often represent more realistic and complex environments. Including such models in the analysis would provide a more comprehensive assessment of the proposed strategies.

- Considering exploration is crucial for addressing the uncertainty in agents' learning dynamics, it would be valuable to investigate how an exploration-focused approach like Random Network Distillation (RND) could impact the results (more than random exploration FETE-RE in Section 6.2). This could provide practical insights into managing uncertainty more effectively in complex multi-agent environments.

---

> ### Author Response · Authors · 2024-11-17
> **Response to Reviewer M5wC (Part 1 of 2)**
>
> We thank the reviewer for the valuable feedback! We have integrated some of your comments into our revision (see discussion in Appx. H). Below, we provide detailed responses to address your concerns and clarify some of your potential misunderstandings of the paper. We hope the revisions address your concerns and encourage you to reconsider the evaluation of our work.
>
> ## Response to Weakness 1
> We are unsure whether we correctly understand your question “_...how understanding the model class $\mathcal{F}$ effectively reduces this uncertainty_”. We interpret it as an inquiry about how the model class $\mathcal{F}$ is chosen in practice. If it is incorrect, could you kindly provide further clarification? Based on this interpretation, we offer the following explanation.
>
> The main principle for selecting $\mathcal{F}$ is to ensure our “realizability” assumption holds with high probability, i.e. the true model $f^* \in \mathcal{F}$. The concrete choice of $\mathcal{F}$ depends on how much prior knowledge is available regarding the agents’ learning dynamics. Generally, the less prior knowledge we have, the larger $\mathcal{F}$ should be to ensure realizability, and vice versa.
>
> In practice, even if $f^*$ is very complex, a “safe-choice” would be treating $f^*$ as a black box and use deep neural networks as the model class $\mathcal{F}$, which have been proven to be strong approximators in deep RL literatures. In our revision in Appx. H, we include this argument, and discuss how our results can be generalized to cases when $\mathcal{F}$ is uncountable but has finite covering numbers (s.t. neural networks can be covered as special cases).
>
> Nonetheless, we emphasize that this paper focuses on the learning process given a model class $\mathcal{F}$, and how to design $\mathcal{F}$ given prior knowledge is an interesting topic but orthogonal to ours.
>
>
> ## Response to Weakness 2
>
> * “_...presents a significant challenge when considering real-world complexities where agents' behaviors are influenced by others”, “... restricts its utility in … agent behaviors are interdependent and continuously evolving…_”
>
>   We believe the reviewer may have some misunderstanding on our setting and results, and we provide further clarification below.
>
>   We highlight, **our framework, particularly the Markovian learning dynamics (Def. 3.1) and model-based setup, already covers a very broad class of scenarios and allow for arbitrarily complex behaviors, including interdependent agents’ policies updates**. Specifically, Def. 3.1 only assumes that the joint policies of all agents ($\pmb\pi_{t+1}$) depends on the last step joint policy ($\pmb\pi_{t}$) and rewards. It does not impose any restrictions on how $\pmb\pi_{t+1}$ is determined by $\pmb\pi_{t}$. Prop. 3.3 and the algorithm designs in Section 5 only require the access to a model class $\mathcal{F}$ under the realizability assumption, and we do not put any restrictions on the structure of $f^*$.
>
>   As noted in our response to Weakness 1, even if $f^*$ is highly complex, it can be just treated as a black box and approximated by neural networks. Neural networks have been extensively validated as powerful function approximators in the supervised learning and deep reinforcement learning literature.
>
>
> * “_...its current design focused on binary or categorical decision-making scenarios_”
>
>   We highlight that, **we focus on the tabular setting, because the steering problem can already become very challenging even in the tabular setting**, as a result of the “curse of multi-agency” (i.e. the complexity of agents’ learning dynamics scales exponentially with the number of agents). When considering the non-tabular setting (i.e. continuous state and action spaces), it becomes much more intractable due to the additional challenge, known as the “curse of dimensionality” (i.e. the complexity of agents’ learning dynamics scales exponentially with the size of state and action spaces).
>
>   Even though, in Appx. H, we use Linear-Quadratic Games as an example, to demonstrate how our current framework can handle non-tabular settings where the policies and steering rewards are parameterized by finite variables. We believe the investigation in this direction is an interesting future direction, but is out of the scope of this paper.
>
>
> * _“...such as, herd behavior or nonconformity”_
>
>   Our steering setting studies how to influence the agents’ learning dynamics by additional steering rewards, which implies, it is specialized for “incentive-driven agents”. The setting with herding or nonconforming agents may not fit into this framework, since the agents may only exhibit limited responsiveness to the reward modification. We believe studying how to steer in those settings is an interesting topic, but is out of the scope of this paper.

---

> ### Author Response · Authors · 2024-11-17
> **Response to Reviewer M5wC (Part 2 of 2)**
>
> ## Response to Question 1
>
> In general, non-Markovian agents setting is intractable, implied by the hardness in learning Partially Observable MDPs (POMDPs). However, we note that our results for Markovian agents can be directly extended to a large class of tractable non-Markovian learning dynamics with finite memory. Specifically, in this setting, the agent's learning dynamics, although not memoryless, depend only on a finite horizon of interaction history. By treating the short memory as the new “state”, our algorithms can be seamlessly adapted to such dynamics. Please refer to our discussion in Appx. H for more details.
>
> Nonetheless, this paper primarily focuses on Markovian agents. As mentioned in the conclusion section, we defer further exploration on non-Markovian settings to future work.
>
>
>
>
> ## Response to Question 2
>
> We would like to clarify a possible misunderstanding by the reviewer. **Our proposed method, FETE (Procedure 2), already considers the exploration efficiency, which is also a key aspect of our contributions**. Specifically, in line 2 of Procedure 2, we optimize the exploration steering strategy $\psi_{Explore}$ to maximize probability of recovering the hidden model from the interaction history by the MLE. While we agree RND could be an interesting direction to investigate in the future, we believe our method provides advantages over RND in terms of theoretical guarantees. This is because it directly uses the success rate of MLE-based model identification as the reward signal to optimize the exploration strategy.
>
>
> Besides, we want to clarify that **FETE is our main algorithm, while FETE-RE (replacing strategic exploration in FETE by random exploration) just serves as an adaptation of the method from the concurrent work [1] to facilitate a fair comparison**. The comparison between our FETE and FETE-RE exactly demonstrate the importance of strategic exploration when exploration challenges exhibit, which is valued by the reviewer.
>
>
> [1] Canyakmaz et al., Steering game dynamics towards desired outcomes

---

> ### Author Response · Authors · 2024-11-22
>
> Dear Reviewer M5wC, thank you again for your time and efforts in the reviewing process! We hope our responses are helpful, and we are happy to answer your further questions if any. If all of your concerns have been addressed, we will appreciate it if you can let us know whether our response changes your evaluation of our work.

---

> ### Author Response · Authors · 2024-11-29
>
> Dear Reviewer M5wC, given that the discussion period is closing to the end, we would really appreciate it if you can let us know if you have further questions. If all of your concerns are addressed, we kindly request you to reconsider the evaluation of our paper.
>
>
> Thank you once again for your efforts!
>
> Best Regards,
>
> Authors

---

> ### Comment · Reviewer_M5wC · 2024-12-03
>
> I sincerely thank the authors for the detailed responses and clarifications. Based on all the replies, I am willing to increase my score.

---

### Meta-Review · Area_Chair_gYF1 · 2024-12-21

**Metareview:**

This paper presents a framework for steering Markovian agents in multi-agent systems using reinforcement learning under model uncertainty, with strong theoretical guarantees and empirical validations. Despite some concerns about scalability to more complex and realistic settings, the work is conceptually novel. It provides a solid foundation for advancing research in this area.

**Additional Comments On Reviewer Discussion:**

Reviewer oazJ raised concerns about extending the framework to uncountable model classes and non-tabular settings. The authors addressed these by incorporating finite covering sets for infinite models and parameterized representations for non-tabular scenarios, with detailed updates provided in Appendix H.

---

### Decision · Program_Chairs · 2025-01-22

Accept (Poster)